# *In vitro* and *in vivo* studies on the activity and selectivity of butoconazole in experimental infection by *Trypanosoma cruzi*

**Gabriela Rodrigues Leite[1], Denise da Gama Jaen Batista[1], Marcos Meuser Batista[1], Krislayne Nunes da Costa[1], Tomás Mac Loughlin[2], Emilia M Barrionuevo[3], Alan Talevi[3], Lucas N Aberca[3], Otacilio C Moreira[4], Amanda Faier-Pereira[4], Beatriz Iandra da Silva Ferreira[4], Maria de Nazaré Correia Soeiro[1]/+**

[1]Fundação Oswaldo Cruz-Fiocruz, Instituto Oswaldo Cruz, Laboratório de Biologia Celular, Rio de Janeiro, RJ, Brasil
[2]Consejo Nacional de Investigaciones Científicas y Técnicas, Universidad Nacional de La Plata, Facultad de Ciencias Exactas, Centro de Investigaciones del Medio Ambiente, La Plata, Buenos Aires, Argentina
[3]Consejo Nacional de Investigaciones Científicas y Técnicas, Universidad Nacional de La Plata, Facultad de Ciencias Exactas, Laboratorio de Investigación y Desarrollo de Bioactivos, La Plata, Buenos Aires, Argentina
[4]Fundação Oswaldo Cruz-Fiocruz, Instituto Oswaldo Cruz, Plataforma de Análises Moleculares, Laboratório de Virologia e Parasitologia Molecular, Rio de Janeiro, RJ, Brasil

**BACKGROUND** The protozoan *Trypanosoma cruzi* causes Chagas disease (CD). There are two drugs available for the treatment with limited efficacy, especially in the later stage. Focusing on drug repurposing by virtual screening of chemical databases, butoconazole (BTZ) was identified as promising hit.

**OBJECTIVES** Our aim was to explore the trypanosomicidal effect of BTZ alone or in combination with benznidazole (BZ) against *T. cruzi*.

**METHODS and FINDINGS** Our *in vitro* assays validated the low cytotoxicity of BTZ and high potency on amastigotes ($EC_{50} = 0.07$ μM), being 24-fold more potent than BZ. Washout assays demonstrated the sterilisation capacity of BTZ, whereas its combination with BZ gave an additive interaction (x$\Sigma$FICI = 0.66). In a mouse model of acute *T. cruzi* infection, BTZ was unable to suppress parasitaemia but ensured the animal survival. BTZ plus BZ reduced parasitaemia and provided higher survival rates than monotherapies. However, quantitative polymerase chain reaction (qPCR) revealed that BTZ + BZ protocol gave 100% of lack of parasitological cure, as parasite satDNA was amplified in the heart of all surviving animals.

**MAIN CONCLUSIONS** Our dataset reinforces the relevance of drug repurposing and combination strategies to advance into the development of novel therapeutic approaches for CD.

Key words: drugs repurpose - Chagas disease - butoconazole - *Trypanosoma cruzi* - drug combination

American trypanosomiasis, also called Chagas disease (CD), is caused by the haemoflagellate protozoan *Trypanosoma cruzi* transmitted through infected triatomine vector insects to several species of mammals.[1] It is considered endemic in 21 Latin American countries, especially in regions with lower human development index, but the identification of CD carriers in non-endemic countries is a concern due to the globalisation and increased migratory flow.[2] According to the World Health Organisation (WHO), more than six million people are infected, with about 30,000 new cases and 12,000 deaths annually due to the development and complications of the disease, representing a relevant global public health problem.[1,2]

CD has an acute phase of short duration (four-eight weeks), usually asymptomatic or oligosymptomatic and with patent parasitaemia. Although parasite proliferation is controlled by a competent host immune response, there is no cure and parasites remain hidden in tissues and organs for years or decades, without developing clinical symptoms. However, this asymptomatic chronic stage may evolve to the symptomatic clinical form, usually cardiac and/or digestive, in about 30-40% of the carriers, which can be debilitating and fatal.[3]

It is well reported that early diagnosis is crucial to increasing the chances of cure, during *T. cruzi* infection. However, most of those affected people only become aware of their diagnosis in the late phase of CD.[4,5]

Financial support: CNPq, FIOCRUZ, FAPERJ, Agencia I+D+I (PICT 2021-0404), PROEP/CNPq/FIOCRUZ.
MNCS and OCM are researchers from CNPq and CNE/FAPERJ.
+ Corresponding author: soeiro@ioc.fiocruz.br
https://orcid.org/0000-0003-0078-6106

In addition, the therapeutic arsenal against CD is unsatisfactory and restricted to the use of two old nitroderivatives: benznidazole (BZ) and nifurtimox (NF). Both require an extended period of administration, display limited efficacy in chronic stages, induce severe side effects, and have limited efficacy against strains of *T. cruzi* that are naturally resistant to these nitroderivatives, factors that favour low adherence and/or discontinuity of the therapy.[6] Based on intensive research on transmission, treatment, and control, the Pan American Health Organisation (PAHO) estimates that among the people affected by CD living in the Americas, only around < 10% can access diagnosis, and of those, only about 1% have access to appropriate care and treatment.[7,8]

Drug repurposing is a promising approach that may combine bioinformatics tools with the availability of drug libraries/chemical databases comprising marketed, discontinued and/or archived drugs, to find a new therapeutic indication, different from the original one.[9] Compared to the process of discovering and developing a *de novo* drug, repurposing has several advantages such as significantly reducing the time and investment costs required in research, bypassing the main barriers that make new advances aimed at neglected tropical diseases, such as CD, unfeasible.[10,11]

Combined therapy has gained more prominence in screenings with successful results in clinical treatment for various diseases.[12] The co-administration of drugs with different mechanisms of action modulates more than one target simultaneously, increasing the chances of improved efficacy with lower doses and reduced time, decreasing adverse effects and the chances of drug resistance.[12,13,14]

The 14α-demethylase (CYP51) inhibitors, indicated to treat fungal infections, have been investigated in preclinical and clinical assays for CD.[15] They act by inhibiting sterol biosynthesis, which is considered as an essential metabolic pathway for membrane remodelling, metabolism, and cell division process of *T. cruzi*.[12] Due to the promising findings in mouse and dog models of *T. cruzi* experimental infection, two azole antifungals advanced into clinical studies in CD chronic carriers.[16,17,18] The CHAGASAZOLE trial aimed to evaluate the efficacy of posaconazole (POSA) in chronic patients with CD. Although the treatment achieved parasitaemia suppression, its efficacy was not sustained after the drug withdrawal.[17,18] STOP-CHAGAS trial (Study of Oral Posaconazole in the Treatment of Chagas Disease) also evaluated POSA in monotherapy regimens and in combination with BZ in asymptomatic chronic patients with CD. The antiparasitic effect of the POSA + BZ combination showed no advantage over BZ monotherapy.[18]

Fosravuconazole (E1224) is a prodrug of ravuconazole (RAV), a CYP51 inhibitor with promising anti-*T. cruzi* results *in vitro* and *in vivo* experimental models.[12] However, E1224 also failed to sustain negative quantitative polymerase chain reaction (qPCR) in a randomised clinical trial with different oral dosing regimens administered to chronic patients.[16] The lack of translation of E1224 and POSA has been attributed to many factors including underdosing and the short treatment period.[6]

On the other hand, the well-known trypanosomicidal activity of CYP51 inhibitors and the metabolic differences on trypanosomatidae and fungi justify the search for novel drugs more selective for the parasite enzymes, as has been reported in a *in vivo* mouse model, reaching high parasitological cure rates especially when combined with BZ,[19] arguing in favour of continuing with studies testing CYP51 inhibitors.[6]

Focusing on drug discovery, Alberca and colleagues conducted virtual screening campaigns based on ligands and target structures and identified butoconazole (BTZ) as a promising anti-*T. cruzi* agent.[20] Then, our aim was to further explore the therapeutic potential of the BTZ *in vitro* and *in vivo* assays of experimental *T. cruzi* infection, in monotherapy or in combination with BZ, both on bloodstream trypomastigotes and intracellular amastigotes, by testing also distinct parasite strains.

## MATERIALS AND METHODS

*Compounds* - BZ (Fig. 1A) obtained from LAFEPE (Pharmaceutical Laboratory of the State of Pernambuco Governador Miguel Arraes). BTZ (Fig. 1B) provided in the form of its nitrate salt by Drs Alan Talevi and Lucas Alberca (Laboratorio de Investigación y Desarrollo de Bioactivos - LIDeB/AR, Buenos Aires, Argentina) based on a previous virtual and *in vitro* preliminary screenings.[20] For our *in vitro* studies, the stock (50 mM) solutions were prepared in dimethyl sulfoxide (DMSO) (Sigma-Aldrich, St. Louis, MO, USA) and used with final working concentrations never exceeding 0.6%, meaning that there is no toxic effect on mammalian cells or parasites *in vitro*.[21] For *in vivo* evaluations, BTZ was dissolved in 10% DMSO while BZ in 3% Tween 80 and then both were diluted in sterile and deionised water.

*Mammalian cell cultures* - L929 fibroblast lineage were cultured in Roswell Park Memorial Institute 1640 medium without phenol red (RPMI - Sigma-Aldrich, St. Louis, MO, USA), supplemented with 10% foetal bovine serum (FBS - Cultilab - SP - Brazil) and 2 mM glutamine (Sigma-Aldrich, St. Louis, MO, USA) with pH 7.2 - 7.4, as previously reported.[21] H9C2, cardiomyoblast cultures, were maintained in Dulbecco's Modified Eagle's Medium (DMEM) high glucose (4500 mg/L) medium supplemented (Sigma-Aldrich, St. Louis, MO, USA) with 10% FBS (Cultilab - SP -Brazil), 2 mM glutamine (Sigma-Aldrich, St. Louis, MO, USA) and 100 μg/mL penicillin/streptomycin (Sigma-Aldrich, St. Louis, MO, USA).[22] Both cell cultures maintained at 37ºC under an atmosphere of 5% $CO_2$, with weekly dissociation protocol using 0.01% trypsin.

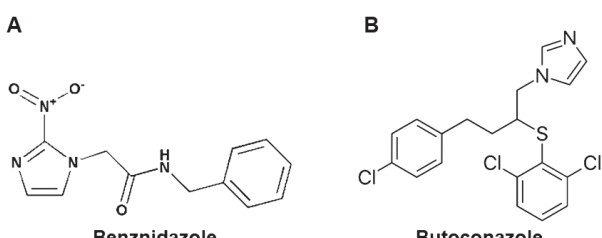

**A**   Benznidazole

**B**   Butoconazole

Fig. 1: chemical structure of the compounds: (A) benznidazole (BZ) and (B) butoconazole (BTZ).

*Parasites* - Trypomastigotes of the Y strain of *T. cruzi* [discrete typing unit (DTU) II] were acquired by cardiac puncture of Swiss Webster mice at the peak of parasitaemia.[23] Trypomastigotes of the Tulahuen strain (DTU VI), expressing the β-galactosidase gene of *Escherichia coli*, were collected from the supernatant of previously infected L929 cultures.[21]

*Cytotoxicity assay* - The toxicity of the compounds was evaluated on two mammalian cell lines (L929 and H9C2), as reported.[21] The cells were seeded in 96-well plates (4 x $10^3$ cells/well - for L929; 25x$10^3$ cells/well - for H9C2) and incubated at 37ºC and 5% $CO_2$. The L929 cultures were kept under treatment for 96 and 168 h with a serial dilution (1:2) of the compounds (up to 400 μM). After the treatment periods, the AlamarBlue® reagent (Thermo Fisher Scientific Inc) was added and incubated for 24 h. The H9C2 cultures were also treated in serial dilution up to 200 μM for 24 h. PrestoBlue® reagent (Thermo Fisher Scientific Inc) was added to the cultures and incubated for 3 h. All plates were analysed in the spectrophotometer SpectraMaxM3 (570 and 600 nm) and $LC_{50}$ values (concentration capable of inducing loss of viability in 50% of the cells) determined.[21] All assays were done at least twice, in triplicates.

*In vitro activity on intracellular forms of T. cruzi* - L929 cells previously seeded in 96-well plates (4 x $10^3$ cells/well) were infected (ratio of 10 parasites:1 host cell) for 2 h with trypomastigote forms (Tulahuen strain transfected with the β-galactosidase gene, DTU VI). After 48 h, the cultures were incubated with the compounds (0 - 10 μM) for 96 h at 37ºC/ 5% $CO_2$ and the enzymatic activity measured at 570 nm after the addition of chlorophenol red (CPRG) to determine $EC_{50}$ and $EC_{90}$ values (minimum concentration able of eliminating 50 and 90% of the infection, respectively).[21]

*In vitro activity on bloodstream trypomastigotes* - Swiss male mice were inoculated with 5 x $10^4$ bloodstream trypomastigotes (BT, Y strain, DTU II) and at the parasitaemia peak [8th post-infection (dpi)], euthanised using 3% isoflurane and blood samples obtained by cardiac puncture (CEUA FIOCRUZ L038-2017). Then, 100 μL of parasite suspension ($10^7$/mL in RPMI medium + 5% FBS) was added to the same volume of RPMI + 5% FBS containing treatment with BTZ (serially diluted - 1:2) at twice the desired final concentration. After 2 and 24 h at 37ºC, the number of live parasites was determined by light microscopy quantification using a Neubauer chamber.[6] Untreated controls were carried out with parasites kept under the same conditions and BZ was run in parallel. The activity of the compounds was expressed by the $EC_{50}$ and $EC_{90}$ values. The assays were done at least twice, in duplicate.

*Drug combination on intracellular forms of T. cruzi* - L929 were seeded in 96-well plates (4 x $10^3$ cells/well) and infected with *T. cruzi (*Tulahuen strain). Predetermined $EC_{50}$ values were used to determine the top concentrations of the test compounds under fixed ratios ensuring that $EC_{50}$ fell in the midpoint of a seven-point two-fold dilution series. The fixed ratios were 5:0, 4:1, 3:2, 2:3, 1:4 and 0:5.

[24,25] According to Odds (2003), the Fractional Inhibitory Concentration Indexes (FICI) was calculated by the ratio of the $EC_{50}$ values of each ratio of the combinations/$EC_{50}$ of each compound alone. Finally, the sum of the FICIs (ΣFICI) is calculated to classify the interaction nature of the combinations.[26] ΣFICI ≤ 0.5 = Synergism; ΣFICI between 0.5 and 4 = additive (No interaction); ΣFICI > 4 = antagonism. At least two independent experiments were performed in triplicate.

*Washout assays in L929 cultures infected with T. cruzi* - L929 cultures were infected with Tulahuen strain as above described and incubated or not for 168 h with 10 μM of BZ and BTZ (37ºC/5% $CO_2$). RPMI culture medium (containing the tested compounds) was replaced every 72 h. Then, the cultures were rinsed with 0.1M PBS, and the drug-free culture medium added for another 168 h, replacing it every 72 h. The number of parasites released into the supernatant and of intracellular forms were analysed by light microscopy and by spectrophotometry (570 nm) after adding CPRG, respectively. The results were expressed as the percentage of reduction in parasite growth compared to untreated cells. BZ was used as a positive treatment control.[25] The data results from two independent experiments done in at least triplicate.

*Ethics* - The pharmacokinetic study that was conducted under supervision of Dr Talevi according to protocols approved by the Institutional Committee for the Care and Use of Laboratory Animals (CICUAL) of the Faculty of Exact Sciences of the University of La Plata, Argentina (FCE-UNLP) (protocol number 004-00-23). In parallel, all animal studies were carried out in strict accordance with the guidelines established by the FIOCRUZ Committee of Ethics for the Use of Animals (CEUA L038-2017), and were approved by CTNBio and CIBIO/IOC/FIOCRUZ for a Biosafety Quality Certificate (CQB 105/99) for the use of GMOs (*T. cruzi* strain Tulahuen transfected with β-galactosidase gene), in addition to registration in the SisGen (A5825BF).

*In vivo pharmacokinetic study* - Adult male specific-pathogen-free BALB/c mice, provided by the Faculty of Veterinary Sciences, National University of La Plata (FCV-UNLP) were used. Animals were housed in cages with four animals per cage, under a 12 h light/dark cycle and controlled temperature, provided with water and food *ad libitum* and environmental enrichment. A minimum period of six days was given after transport for physiological acclimatisation before any procedure. Five groups of mice (n = 3 per group) were used, one per sampling time. A fresh solution of BTZ at 50 mg/mL in DMSO (BIOPACK) was administered intraperitoneally (i.p.) at an injection volume of 2 mL/kg, achieving a final dose of 100 mg/kg. Each mouse was sampled only once, collecting a maximum of 150 μL of submandibular blood at 1, 2, 4, 8 and 24 h post-administration, and transferred into EDTA-treated tubes. Plasma was obtained immediately after blood collection by centrifugation for 10 min at 4,000 rpm and conserved at -17ºC until quantification. After the experiment, all animals were euthanised using $CO_2$. The i.p. route of BTZ was chosen due to its low oral bioavailability.

*Plasma sample preparation and LC-MS/MS analysis* - The plasma samples were treated by keeping them in an ice-water bath. An equal volume of cold acetonitrile (pre-cooled in a freezer) was added to induce protein precipitation. The mixture was vortexed for 10 s and then allowed to stand for 15 min in the cold bath. Subsequently, the samples were centrifuged at 10,000 rpm at 4ºC for 15 min. After, the supernatant was collected, diluted with water/acetonitrile (1:1), and filtered through a 0.22 μm nylon membrane filter before being transferred to an insert for injection into the LC-MS/MS system.

The equipment used was a Waters Alliance 2695 HPLC coupled to a tandem quadrupole mass spectrometer, Premier XE (Waters Corp.), equipped with an electrospray ionisation source operating in positive mode (ESI+). Chromatographic separation was performed using an X-SELECT™ $C_{18}$ column (75 mm × 4.6 mm, 3 μm pore size; Waters Corp.) with a gradient of methanol and ultrapure water, with formic acid and ammonium acetate as ionisation additives. The mass spectrometer was configured to operate in multiple reaction monitoring (MRM) mode, employing two mass transitions for the compound, one for quantification (411.0 > 164.9) and the other for confirmation (413.2 > 164.9).

Unbound drug concentrations in plasma were estimated by multiplying the analytically determined total drug concentrations by the unbound fraction predicted by Deep-PK.[27]

*In vivo analysis in a mouse model of acute T. cruzi infection* - Swiss male mice (18 - 20 g) were obtained from the Institute of Sciences and Technologies in Biomodels (ICTB-FIOCRUZ), housed with a maximum of 5 animals per cage and kept in a specific pathogen-free (SPF) room at 20 - 24ºC under a 12 h light/dark cycle. All animals received sterilised water and food *ad libitum*. The animals were acclimatised for seven days before the experiments. For a proof-of-concept study, animals were infected by i.p ($1x10^4$) with bloodstream trypomastigotes (Y strain). The drugs administration started at the onset of parasitaemia (5th day dpi), only using mice with detectable parasitaemia.[19,28] Control mice groups were age-matched and housed under identical conditions. The following experimental groups were used ≥ 3 mice per group: untreated (infected vehicle-treated control) and treated orally (p.o) with BZ (10, 25 and 100 mg/kg/day), corresponding to suboptimal and optimal doses of the reference drug; infected and treated with BTZ (i.p) using concentrations adjusted to be equimolar to BZ doses (18.24, 45.61 and 182.44 mg/kg/day). Also, the following combo was administered: BZ (10 mg/kg/day/p.o) + BTZ (18.24 mg/kg/day/i.p). The animals received the drugs once a day for five consecutive days. A 10-fold lower optimal dose of BZ (10 mg/kg/day) was chosen to clearly observe any benefit of its association with BTZ. The i.p. route for BTZ was chosen due to its low aqueous solubility.

*Parasitaemia and mortality rate* - Parasitaemia was performed by collecting 5 μL of the blood from the tail's vein, disposing it on a slide and covering with coverslips (18 x 18 mm) for direct counting of parasites observed in 50 fields under light microscope (40X magnification).[19,28] The mortality was followed daily up to 30 days after the end of treatment and expressed as a percentage of cumulative mortality (% CM).[19,27] I*n vivo* assays were conducted with n = 5 per group.

*In vivo efficacy assessments by qPCR evaluations* - At the end point, the animals were weighed; euthanised and submitted to cardiac puncture. 500 μL of fresh blood was collected and added to 1 mL of guanidine. The DNA was extracted from 300 μL of guanidine-EDTA blood, using the kit High Pure PCR Template Preparation (Roche diagnostics, Mannhein, Germany), according to the manufacturer's instructions. At the last step of the protocol, DNA was eluted in 100 μL of elution buffer. Also, cardiac tissues were collected, weighed and fragmented using a scalpel, 200 μL of Tissue Lysis Buffer was added and samples immediately placed in Styrofoam with dry ice. Blood and heart tissues were stored in the freezer at -80ºC until analysis by qPCR. Heart tissues were homogenised using a Tissue Homogeneizer (Qiagen). DNA was extracted using the kit High Pure PCR Template Preparation (Roche diagnostics, Mannhein, Germany), according to the manufacturer's instructions for tissue samples. qPCR was done using a TaqMan multiplex assay, targeting *T. cruzi* satDNA and mouse GAPDH, as an endogenous control, and to normalise the parasite load. To obtain standard curves, serial dilutions of the DNA were prepared in appropriate plates for the assay. After normalising the results obtained on the samples, the parasite load was expressed as parasite equivalents/mL when evaluated in blood, or parasite equivalents/mg when in mouse tissue. In all trials, positive and negative controls were also tested in parallel.

*Data analysis and statistics* - The data were plotted and evaluated using the GraphPad Prism version 5.0 program (GraphPad Software, San Diego, CA, USA). The Deep-PK platform (available for free at https://biosig.lab.uq.edu.au/deeppk) was applied for prediction of PK and toxicity data of BTZ.[27]

## RESULTS

The first step was evaluating the cytotoxic profile of BTZ on L929 cells treated for 96 h. BTZ displayed a $LC_{50}$ value of 23.9 μM ± 1.1. while BZ was non-toxic up to 400 μM (Table I, Fig. 2). Next, the trypanosomicidal activity was investigated using L929 cultures infected with the Tulahuen strain (Fig. 3A-B, Table I). BTZ was highly potent with $EC_{50}$ = 0.07 ± 0.02 μM while BZ reached 1.73 ± 0.16 μM. Regarding selectivity, BTZ displayed an excellent SI of 341, being about 24-fold more potent than the reference drug (BZ) (Fig. 3, Table I).

To further explore toxicity of BTZ *in vitro* now upon longer exposure, L929 were treated for 168 h. Our findings showed that BTZ and BZ displayed $LC_{50}$ values of 17.53 μM ± 3.24 μM and > 200 μM, respectively (Table I).

The potential cardiotoxic effect was also assessed using H9C2 cell lines. Our findings demonstrated $LC_{50}$ values of 36 ± 0.25 and >200 μM after 24 h of incubation with BTZ and BZ, respectively (Table II).

TABLE I

The activity (mean value and standard deviation - SD) of benznidazole (BZ) and butoconazole (BTZ) against intracellular forms (Tulahuen strain) of *Trypanosoma cruzi*, the mammalian cytotoxicity and respective selectivity indexes (SI)

| Compounds | Activity against *T. cruzi* intracellular forms - µM - 96 h (pEC$_{50}$) | | Cytotoxicity on L929 cells - µM - 96 h (pEC$_{50}$) | Cytotoxicity on L929 cells -µM - 168 h (pEC$_{50}$) | |
|---|---|---|---|---|---|
| | EC$_{50}$[a] | EC$_{90}$[a] | LC$_{50}$[b] | LC$_{50}$[b] | SI[c] |
| BZ | 1.73 ± 0.16 (pEC$_{50}$ 5.76) | 7.65 ± 1.69 | > 400 | > 200 (pEC$_{50}$ 3.69) | > 231 |
| BTZ | 0.070 ± 0.02 (pEC$_{50}$7.15) | 3.88 ± 0.45 | 23.90 ± 1.1 | 17.53 ± 3.24 (pEC$_{50}$4.62) | 341 |

*a*: EC$_{50}$/EC$_{90}$ concentration capable of reducing 50% and 90% of infection, respectively; *b*: LC$_{50}$ concentration capable of reducing 50% of viable cells; *c*: SI ratio between LC$_{50}$/EC$_{50}$ values considering 96 h of incubation.

The effect against bloodstream trypomastigotes (BT) revealed that BTZ was 2.3-fold less potent than BZ after 24 h of incubation displaying EC$_{50}$ values of 13.83 ± 0.70 and 5.93 ± 0.15 µM, respectively (Table II).

Washout assays showed that BTZ and BZ suppressed the parasite release into the supernatant of L929. Also, BTZ and BZ incubation resulted in > 96% reduction of the intracellular parasitism (Table III).

The BTZ+BZ combo effect against intracellular forms demonstrated that the FICI ranged from 0.35 to 1.28 revealing an additive drug interaction, with a value xΣFICI = 0.66 (Table IV, Fig. 4).

The next step was to evaluate the concentration vs. time profile of the compound in mice after administration in a single dose of 100 mg/kg of BTZ. The plasma drug concentration levels were monitored at 1, 2, 4, 8 and 24 h after administration (Fig. 5). The mean peak concentration after 2 h of administration showed a total plasma concentration of 25.9 µM. Interestingly, when correcting the total concentration by the estimated unbound fraction calculated with Deep-PK (1.67%), even a single administration with this dose would provide an unbound concentration above the EC$_{50}$ against amastigotes for around 7 h. Noteworthy, since the estimated EC$_{50}$ considers both bound and unbound drug in the culture medium, the unbound EC$_{50}$ value is expected to be below the observed EC$_{50}$ if binding to proteins in the culture medium has occurred.

Based on the compilation of our findings, we proceeded with the antiparasitic activity of BTZ tested *in vivo* in a mouse model of acute infection. The animals were treated alone or in co-administration with BZ. Male Swiss mice were infected with *T. cruzi* and treat-

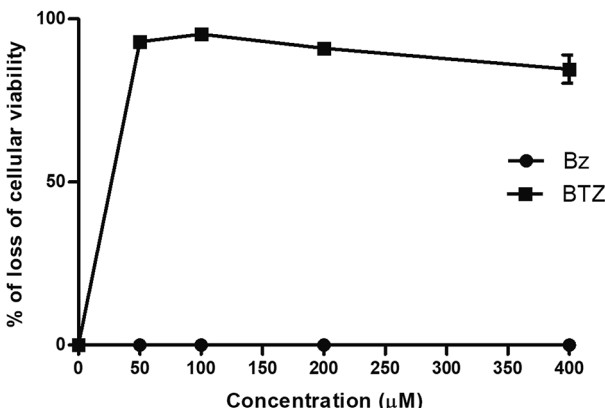

Fig. 2: cytotoxicity profile of benznidazole (BZ) and butoconazole (BTZ) on L929 mammalian cell lines after 96 h of incubation at 37ºC.

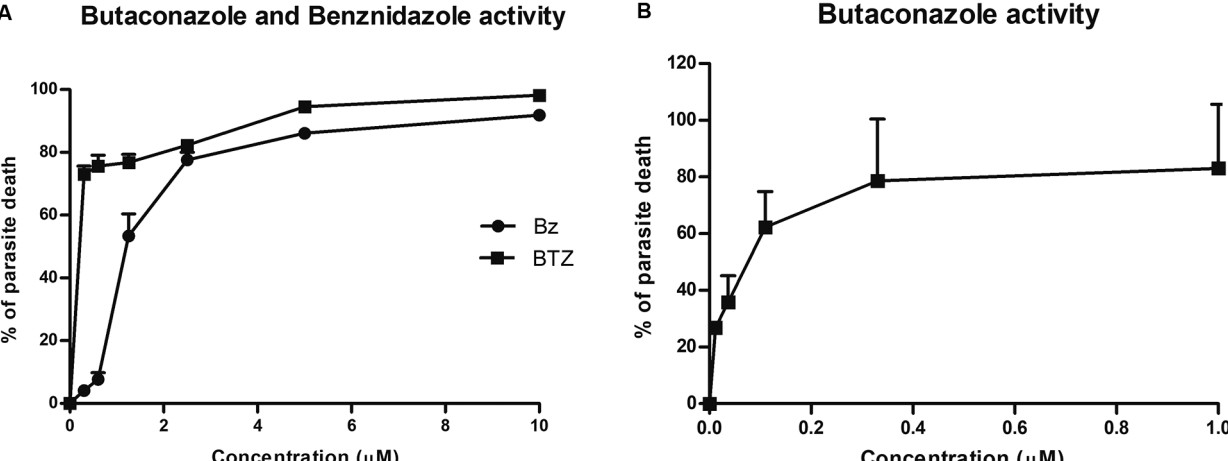

Fig. 3: the trypanosomicidal effect of butaconazole (BTZ) and benznidazole (BZ) against intracellular forms of *Trypanosoma cruzi* (Tulahuen strain) after 96 h of incubation at 37ºC. (A) Dose-dependent activity of BZ and BTZ up to 10 µM. (B) Dose-dependent activity of BTZ up to 1 µM.

TABLE II

The activity (mean value and standard deviation - SD) of benznidazole (BZ) and butoconazole (BTZ) against bloodstream trypomastigotes (Y strain) of *Trypanosoma cruzi*, their cardiotoxicity and respective selectivity indexes

| Compounds | Activity on *T. cruzi* (Y strain) blood trypomastigotes - μM (pEC$_{50}$) | | | | Toxicity on H9C2 mammalian cells in μM (pEC$_{50}$) |
|---|---|---|---|---|---|
| | EC$_{50}{}^a$ - 2 h | EC$_{90}{}^a$ - 2 h | EC$_{50}{}^a$ - 24 h | EC$_{90}{}^a$ - 24 h | LC$_{50}{}^b$ - 24 h |
| BZ | > 20 | > 20 | 5.93 ± 0.15 (pEC$_{50}$ 5.22) | 17.54 ± 0.65 | > 200 (pEC$_{50}$ 3.69) |
| BTZ | > 20 | > 20 | 13.83 ± 0.70 (pEC$_{50}$ 4.85) | 18.64 ± 0.11 | 36.36 ± 0.25 (pEC$_{50}$ 4.43) |

*a*: EC$_{50}$/EC$_{90}$ concentration capable of reducing 50% and 90% of parasites, respectively; *b*: LC$_{50}$ concentration capable of reducing 50% of viable cells.

TABLE III

The washout analysis performed against intracellular forms of *Trypanosoma cruzi* (Tulahuen strain) in L929 cell lines after incubation with benznidazole (BZ) and butoconazole (BTZ). The data displays the percentage (%) of reduction in the number of released and internalized parasites. Results represent mean and standard deviation (SD)

| Compounds | Released parasites | Intracellular forms |
|---|---|---|
| BZ | 100 ± 0 | 98.6 ± 1.4 |
| BTZ | 100 ± 0 | 96 ± 4 |

ed for five consecutive days, starting at the parasitaemia onset (5 dpi). Two different assays were performed. In the first one, animals were treated with 25 and 45.61 mg/kg/day of BZ and BTZ, respectively. Our findings demonstrated that BZ and BTZ gave 93 and 50% of peak decline at the 8 dpi (Fig. 6A). Regarding mortality, while untreated mice reached 80% of death, BZ and BTZ gave 0 and 40%, respectively (Fig. 6B).

Next, sub-optimal and optimal doses of both drugs (10 and 100 mg/kg/day for BZ and 18.24 and 182.44 for BTZ, respectively) were tested. BTZ at its highest concentration (182.44 mg/kg/day) reduced by 63% the parasitaemia (p = 0.45), while BZ (100 mg/kg/day) suppressed the peak (p = 0.000047) (Fig. 6C).

The lowest concentrations of both BTZ and BZ only slightly (p > 0.05) reduced the parasitaemia (Fig. 6B). Regarding animal mortality, at the lower doses, BTZ and BZ reached 75% and 100% mortality (Fig. 6D). As depicted in the Fig. 6C, the combo treatment (BZ + BTZ) reduced by 47% (p = 0.026) the parasitaemia with higher effect as compared to the drugs given alone (Fig. 6C). The combo and the higher doses of BTZ and BZ resulted in 100% animal survival, contrasting to 33% of mice survival in vehicle group (Fig. 6D).

The qPCR analysis on blood samples from the first trial showed that all surviving mice treated with BTZ (45.61 mg/kg/day) were negative (three of three animals), while for the BZ group, parasite DNA was amplified in all animals (Fig. 7A). In the second trial, regarding blood samples, only one animal treated with BZ and BTZ, at higher doses, amplified parasite DNA. All surviving animals in the COMBO group were neg-

ative for parasite DNA amplification (Fig. 7B). When cardiac samples were evaluated, BZ 100 mg/kg/day gave 100% negative qPCR, while only two of five mice treated with BTZ 184 mg/kg/day were negative. For COMBO (BTZ + BZ, at 18.24 and 10 mg/kg/day, respectively), all surviving mice showed positive amplification for parasite DNA, demonstrating drug failure (data not shown).

## DISCUSSION

The current available drugs to treat CD have unsatisfactory efficacy and several limitations, reinforcing the need for new low-cost treatments that have higher safety and efficacy.[6,12,29]

The clinical success of repurposing drugs has stimulated the systematic approach for drug screening, providing the potential to reduce costs and time between preclinical and clinical steps, which are especially valuable characteristics to address the development of new therapeutics solutions for neglected tropical diseases.[30] Among the treatment strategies, the drug combinations show promising results in different therapeutic fields resulting in increased efficacy, reduced dose regimens and time of drug administration that minimises the chances of adverse effects and drug resistance, enhancing patient adherence.[6,31]

Therefore, the objective of our study was to evaluate the therapeutic efficacy of BTZ alone or in combination with BZ on experimental *T. cruzi* infection. BTZ was identified as a promising drug candidate for CD by Alberca and colleagues using a computational strategy to explore repurposing-oriented chemical databases.[20]

Our present data corroborates these studies regarding the trypanosomicidal effect of BTZ against *T. cruzi in vitro* as it was highly effective against intracellular amastigotes at sub micromolar doses, being about 24-fold more potent than BZ. However, BTZ was less potent against bloodstream trypomastigotes as compared to the reference drug. Our data are aligned to the current literature related to the trypanosomicidal effect of sterol biosynthesis inhibitors, especially against the multiplicative amastigote forms, which are much more metabolically active than trypomastigotes.[31,32,33] In parallel, cytotoxicity tests were conducted on different mammalian cell types, and a mild time-dependent toxicity was observed, as reported besides a mild cardiotoxicity.[20]

TABLE IV

The analysis of the combinatory effect of butoconazole (BTZ) and benznidazole (BZ)
on intracellular forms of *Trypanosoma cruzi* (Tulahuen strain) in L929 cell lines

| Proportion of each drug | Drug concentration | | FICI[a] | | | |
|---|---|---|---|---|---|---|
| BTZ \| BZ | BTZ | BZ | BTZ | BZ | ∑FICI[b] | x∑FICI[c] |
| 5 \| 0 | 0.031 | - | 1 | 0 | | |
| 4 \| 1 | 0.024 | 0.738 | 0.76 | 0.51 | 1.28 | |
| 3 \| 2 | 0.006 | 0.402 | 0.20 | 0.28 | 0.48 | |
| 2 \| 3 | 0.012 | 0.219 | 0.37 | 0.15 | 0.52 | 0.66 |
| 1 \| 4 | 0.008 | 0.138 | 0.26 | 0.10 | 0.36 | |
| 0 \| 5 | - | 1.445 | 0 | 1 | | |

*a*: fractional inhibitory concentration index; *b*: sum of the FICIs; *c*: mean value of ∑FICIs.

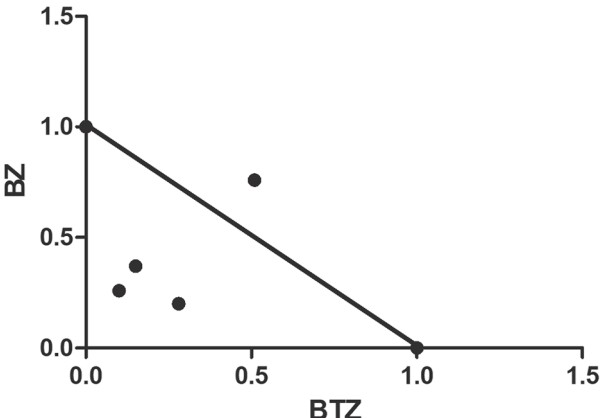

Fig. 4: isobologram of benznidazole (BZ)and butoconazole (BTZ) used in combination. The result shows an additive effect on intracellular forms of *Trypanosoma cruzi* (Tulahuen strain).

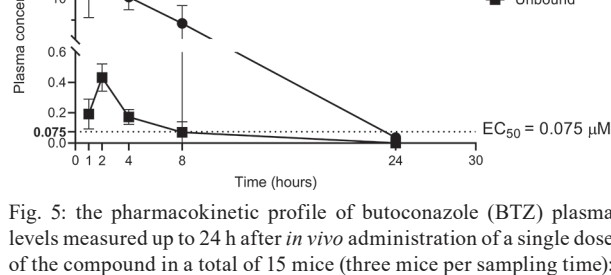

Fig. 5: the pharmacokinetic profile of butoconazole (BTZ) plasma levels measured up to 24 h after *in vivo* administration of a single dose of the compound in a total of 15 mice (three mice per sampling time): (A) Mean total plasma concentration - where the peak recorded was 25.9 µM in the first 2 h after administration; (B) Estimated unbound drug concentration vs. time. The unbound drug concentration was estimated using Deep-PK tool to predict the unbound fraction (1.67%) and revealed that BTZ, even in a single dose, has a high potential to provide unbound plasma concentrations above $EC_{50}$ for at least 7 h.

One of the main obstacles to be overcome for the successful treatment of CD is to find an alternative that is effective and sustained in both phases of the disease avoiding parasite relapses.[6] Therefore, we performed washout assays and monitored the parasite release into the cell culture supernatant besides evaluating the percentage of intracellular parasitism to inspect for quiescent parasites. BZ and BTZ eliminated parasite release into the supernatant but were unable to sterilise the infection, achieving about 96 - 98% of decline in the intracellular parasite load due to presence of latent forms.

Another approach was evaluating the effect of BTZ + BZ association *in vitro*. Our findings demonstrate an additive effect (x∑FICI = 0.66), remarkably close to a synergistic interaction, which is a very promising aspect.

The PK study demonstrated that a single 100 mg/kg dose of BTZ, i.p, was readily bioavailable, with peak plasma concentrations two hours after administration and, more important, estimated unbound concentrations above the $EC_{50}$ against amastigotes over several hours. Reasonably, the exposure is expected to be higher in the context of repeated dosing schemes or upon the administration of higher doses.

In view of these results, we checked a proof-of-concept using a murine model of experimental acute infection. BTZ administered alone gave moderate trypanosomicidal activity, achieving a maximum reduction of 63% at the peak when the highest concentration was administered (182.44 mg/kg/day). BZ (100 mg/kg/day) fully suppressed parasitaemia and gave 100% animal survival.

The lowest dose of BTZ (18.24 mg/kg/day) guaranteed mice survival ≥ 50%, while in the lowest dose of BZ (10 mg/kg/day) 100% animal mortality was observed. Also, the combination of the lowest dose of BZ and BTZ reaching 47% of parasitaemia decline, improved the effect *in vivo* as compared to monotherapies providing 100% of mice survival. This data corroborates the promising *in vitro* additive activity profile.

The qPCR findings of mouse blood and heart samples demonstrated that the cardiac tissues provide a more sensitive detection of *T. cruzi* infection. Although 75% of the BTZ (182.44 mg/kg/day) gave negative amplification in all blood samples, the corresponding cardiac analysis demonstrated cure failure in 50% of the surviving animals. Also, in the combo group composed by lower doses of both drugs, despite all treated mice

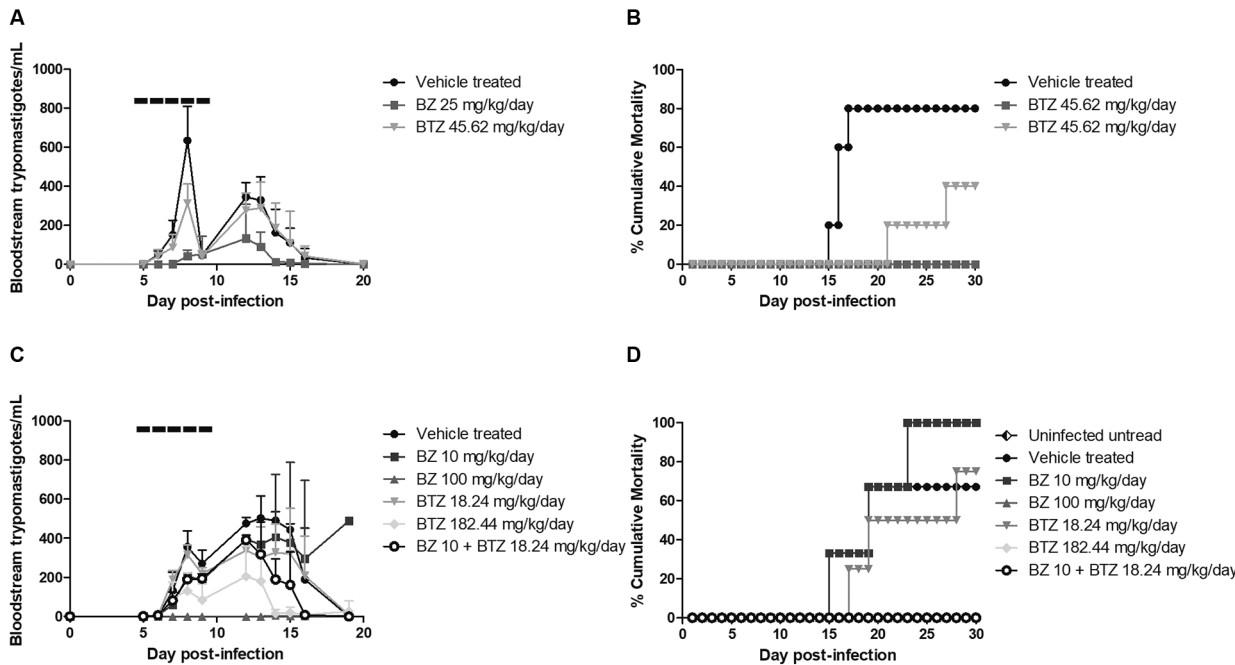

Fig. 6: *in vivo* effect of benznidazole (BZ) and butoconazole (BTZ) alone (A and B) or in combination (C and D) on experimental mouse model of *Trypanosoma cruzi* infection. (A and C) Parasitaemia curves from assays 1 and 2 respectively; (B and D) Percentage of cumulative death (%) relative of assays 1 and 2, respectively. Five mice (n = 5) were used in each experimental group. Dashed lines (-----) mean the period of drug administration (five consecutive days - 5-9 dpi).

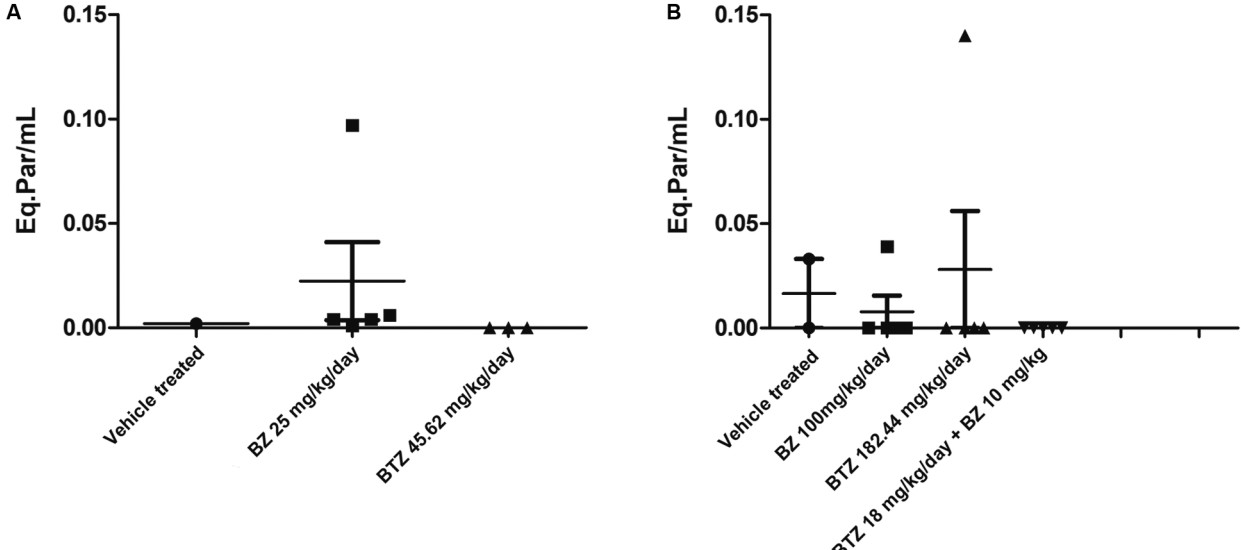

Fig. 7: the determination of blood parasite load by quantitative polymerase chain reaction (qPCR) in benznidazole (BZ) and butoconazole (BTZ) treated mice alone (A) or in combination (B) during *Trypanosoma cruzi* infection. (A) vehicle, BZ (25 mg/kg/day) and BTZ (45.62 mg/kg/day) treated animals (assay 1) and (B) vehicle, BZ (100 mg/kg/day) and BTZ (182.44 mg/kg/day), and combo (BZ + BTZ (assay 2).

survived, no parasitological cure was achieved since the parasite satDNA was amplified in the heart of all surviving animals.

The bulk of our findings encourage future studies performing different combo protocols with these as well as other repurposing drugs to contribute for new therapeutic options for treating neglected tropical diseases.

*In conclusion* - Our findings corroborated and expanded previous findings regarding drug repurpose and combination approaches for experimental analysis of novel therapies for CD. Further studies using additional protocols as well as BTZ analogues may contribute to the identification of novel therapeutic strategies to combat this silent and neglected disease.

## ACKNOWLEDGEMENTS

To Dr Manuel Flores for his assistance with sample preparation.

## AUTHORS' CONTRIBUTION

GL - Contributed with the *in vitro* and *in vivo* assays and data analysis, writing and editing the manuscript; AT - financial support, methodology and data analysis, and manuscript review; LA - provided the compound (butoconazole) and data analysis; MEB - performed the *in vivo* PK study; TM - performed the analytical assays of blood samples extracted during the PK study; OCM, AFB and BISF - methodology and qPCR analysis; MNCS - financial support, project and methodology design and data analysis, edits, and manuscript review. The authors declare no conflict of interest.

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

# OPEN PEER REVIEW

Memórias do IOC thanks the anonymous reviewers for their contribution to the peer review of this work.

## FIRST REVIEW ROUND

REVIEWERS' COMMENTS

### REVIEWER #1

This manuscript presents interesting and relevant results in the field, as it is a comprehensive study that evaluates the in vitro and in vivo effects of the CYP51 inhibitor BTZ against T. cruzi, in addition to its pharmacokinetic profile. The study is conducted with a particular focus on the efficacy of BTZ when used as a monotherapy or in combination with other agents. While the results were not as promising as one might have hoped, I still believe that the work has merit for publication in Memórias do Instituto Oswaldo Cruz. Nevertheless, certain aspects of the study require further attention before it can be accepted for publication. These are outlined below.

The term "more active" should be replaced with "more potent." For example, the statement "23-fold more active than BZ" in lines 39 and 40 requires clarification. A review of all similar cases within the manuscript is required.

Line 70: The term "anti-tripanosomicidal" is erroneous. The suffix "cidal" denotes the act of causing death; thus, the prefix "anti-" is not applicable (e.g., bactericidal vs. anti-bactericidal). The correct form is "trypanosomicidal" or "trypanocidal" (the latter is preferable).

The rationale behind the administration of BTZ via the IP route is unclear, given that the oral route is typically preferred for the development of antichagasic drugs. It is understood that the oral bioavailability of the compound in question is likely to be low due to its low aqueous solubility. However, this information should be presented in the article for the benefit of the reader.

In lines 241 to 243, it is stated that "A fresh solution of BTZ 100 mg/kg diluted in DMSO (2mL/kg, BIOPACK) was administered intraperitoneally (i.p.)." However, the mg/kg unit is used for dose, not for concentration in solution. It appears that there has been a misunderstanding between the units used to express concentration and dose. This should be corrected.

It would be beneficial to present the in vitro inhibition curve for BTZ, particularly to ascertain the extent of its maximum inhibition. CYP51 inhibitors typically exhibit a greater trypanostatic than trypanocidal effect, which may contribute to the observed lack of efficacy of BTZ in this study, particularly when considering the Cmax observed in the PK study.

Line 335: It is stated that the results are in Table I, yet this is not included in the file that was sent. In fact, none of the tables that were referenced are present; only the figures are presented at the end of the manuscript.

### REVIEWER #2

a) Adequacy of the abstract;
Adequate. However, significant improvements must be made to the results to consider them adequate for publication.

b) Originality and importance of the contribution for the development of the field of study
The article's proposition will contribute to the field of neglected diseases drug discovery since the authors performed pharmacokinetics analysis - which still is overlooked by most of the academic research groups - in addition to in vitro potency with very good methods and carry out interesting in vivo assays. Nonetheless, the authors could present a better manuscript by taking more advantage of the several techniques used and presenting poor data or not presented at all, considering that five of nine results are tables not provided by the authors.

c) Methodology, results and discussion;
All results the authors intend to show as tables must be demonstrated as graphs because tables are unreliable for verifying the quality and robustness of the intended results. Tables could be maintained to summarize the results better.

The authors suppress entirely the error values in one of the most important results, the parasitemia graphs. It is known that Pizzi-Brenner quantification in mice models, particularly in non-isogenic lineages such as Swiss mice, generates highly variable data. Therefore, it is only acceptable to show these results with the error values.

Considerations about experiment design: The authors use several treatment regimens, leading to several groups with a few mice per group. Thus, the robustness of the data was decreased, especially considering the high intrinsic variance of the method chosen to access the parasite load. No statistical analysis was provided in the text, or the graphs and statements were made based on weak data.

The statement content between lines 69 and 71 needs to be clarified.

Separate numbers from units in all text.

Detailed information such as strain, DTU, and abbreviation should be removed from the subtitle and included in the text to clarify the main goal of the assay

Despite several assays using different mammalian cell lines and parasite strains, the authors did not show any graph results regarding the cytotoxicity in vitro data.

Provide the cytotoxicity and anti-trypanocide activity data as a fit curve graph. Since GraphPad analysis provides a value of EC50 in almost all cases, even when the curve is not properly fitted with upper and bottom plateus, the EC50 value by itself is unreliable. Considering that the authors indicate in material and methods that the assays were performed at least twice, the EC50 must demonstrate the SD between those assays.

To ensure a compatible comparison of data obtained in this study with others in the literature, please inform EC50 obtained for BZ in all assays in the text, as much as it was informed for the test compound.

To make it easier and straightforward, the comparison with other parameters used in drug discovery and data integration by software used in the field of medicinal chemistry has used EC50 values as pEC50 (pEC50 = –Log (EC50 [M])). Therefore, include the pEC50 values in the tables along with EC50.

Regarding the anti-parasitic assays, the range used to test the parasites with BZ (as control) from 0 to 10uM is unlikely to reach the inhibition plateau needed to properly calculate EC50 since 100% (or nearly) was probably achieved. Please provide the graphs demonstrating efficacy concentrations. Two articles that can instruct the authors to provide this graph according to pharmacology guidelines:

Neubig RR, Spedding M, Kenakin T, Christopoulos A; International Union of Pharmacology Committee on Receptor Nomenclature and Drug Classification. International Union of Pharmacology Committee on Receptor Nomenclature and Drug Classification. XXXVIII. Update on terms and symbols in quantitative pharmacology. Pharmacol Rev. 2003 Dec;55(4):597-606. doi: 10.1124/pr.55.4.4. PMID: 14657418.

Sebaugh JL. Guidelines for accurate EC50/IC50 estimation. Pharm Stat. 2011 Mar-Apr;10(2):128-34. doi: 10.1002/pst.426. PMID: 22328315.

Please, provide complete information about the source of bloodstream trypomastigotes in the section about in vitro bloodstream trypomastigotes assay, including host species, initial inoculum, day of infection, method of blood puncture, anesthesia method, and ethics committee protocol approval number.

Authors mention in line 209 seven dilutions, including one of 1:2; however, in line 204/205, six dilutions were mentioned, and none of them are 1:2 factor. Please rewrite the paragraph between lines 202 and 213 to improve reader comprehension.

Line 204/205: Include the concentration values in uM since, traditionally, the isobologram is a graph of concentrations.

Line 275: Specify the value of inoculum to avoid any misled interpretation. (ie: 1x10^4 or 5x10^4)

Can the authors explain the chosen suboptimal concentrations of BZ for combo treatment? BZ 10 mg/kg/ for 5 days in the acute stage was already proven to have little contribution to treatment outcome due to extremely low blood exposure.

To improve animal well-being, mortality rate assays are strongly discouraged from being performed without humanized end points guided by clinical and behavioral assessments such as body temperature, lack of movement rates, and grimace scale to measure rodent pain. Did the authors apply any of these measures to secure experiment refinement?

Until line 299, no reagent supplier was informed, except by tested compounds. Please provide all suppliers for all commercial reagents used to ensure text uniformity and experimental reproducibility. Place of origin is not necessary.

Please move the ethics statement before the description of the in vivo methods.

Lines 354 to 360 belong to the description of the methods and should not be placed in the results section.

The authors cannot state that the death of BTZ is due to parasitic infection since they did not include a group treated and non-infected, and no information about toxicity was provided. If it is not possible to perform the experiment including this group, data from the literature is useful since it is a drug already approved. However, authors must provide information with similar models and treatment regimens regarding mice strain, age, BTZ dose, and route.

Refine the text to improve readers' comprehension. Describe the treatment regimen instead of more generic terms. ie: BZ optimal dose for BZ 100 mg/kg.

qPCR results: Improve text consistency, providing the results as positive/total or negative/total. ie: 3/3, 0/4.

Since qPCR is a quantitative technique, the authors can explain the reason why they chose to show the data as a table instead of plotting them as graphs exhibiting the quantified amount of DNA of each group.

d) References;

Please revise the references addressing the statements in the article or websites that the main contributions are the subject of the information stated in the manuscript. For instance, but not restricted to these examples:

The reference of the DNDi website indeed contains the referred information; however, studies that properly assess this knowledge should be referred to instead, along with the factsheets of health and research organizations.

References 4,5, and 6 contain information about Chagas disease prevalence. However, the main subjects covered by references 4 and 5 encompass other knowledge areas, and the referred statements are only part of the introduction.

Revise all references in sequences since they are presented in different styles—for instance, line 102 and line 106.

Line 107 and 108 -  Instead of citing more wide reviews as references 14, 25, 26, 27, 28, use other articles that discuss more deeply and are the original source of scientific knowledge about CYP51 inhibitors as treatment of CD. Including them in the introduction can bring a more refined discussion. This action intends to address the proper researcher recognition and avoid the propagation of misleading data interpretation.

Citation error: Revise the reference number 16 because the statement between lines 133 and 135 does not match the article content. The reference the author probably wanted to cite is number 18 instead of No 16, and this error was noticed more than once. Please revise all citation orders; since other misled occurrences were noticed, such as reference number 31; from this point forward, it is not possible to verify the reference information.

e) Figures and tables.

Figure 2: Why did the authors plot the isobologram with segmenting for both axes since the concentration is linear?

The data of 5:0 and 5:0 is not plotted in the graph.

Show axis titles with complete descriptions of units.

Figure 3: Provide only one graph with the total and unbound fraction data. Include in the graph as 'cutoff' the EC50 value. Despite this, it is important to consider that the EC50 alone is not a reliable value since it should be considered the protein binding capacity in the culture medium.

Figure 4: There are no error bars in any parasitemia data, which is impossible since several mice were used in this experiment. Statistical analysis is not shown or no difference was observed in the data.

The reason for the separated graphs in A and C needs to be clarified.

In the axis title, use complete and precise words. Use "trypomastigotes" instead of "par"

Include the treatment period in the graphs as a shaded area or dotted lines to represent the treatment.

Revise the legend's information carefully, aiming to be precise and more informative.  It presents errors in the inoculum, and non-informative/ redundant statements were made.

Please see the pattern of legends used for research groups experienced in Chagas disease proof-of-concept assays.

Francisco AF, Jayawardhana S, Lewis MD, White KL, Shackleford DM, Chen G, Saunders J, Osuna-Cabello M, Read KD, Charman SA, Chatelain E, Kelly JM. Nitroheterocyclic drugs cure experimental Trypanosoma cruzi infections more effectively in the chronic stage than in the acute stage. Sci Rep. 2016 Oct 17;6:35351. doi: 10.1038/srep35351. PMID: 27748443; PMCID: PMC5066210.

## AUTHORS' RESPONSE TO THE REVIEWERS

Dear Editor,

Please, find attached the revised manuscript entitled "In vitro and in vivo studies on the activity and selectivity of butoconazole in experimental infection by Trypanosoma cruzi" by Gabriela Rodrigues Leite, Denise da Gama Jaen Batista, Marcos Meuser Batista, Krislayne Nunes da Costa, Tomás Mac Loughlin, Emilia M. Barrionuevo, Alan Talevi, Lucas N. Aberca, Otacilio C. Moreira, Amanda Faier-Pereira, Beatriz Iandra da Silva Ferreira, and Maria de Nazaré Correia Soeiro to be considered for publication in MEIOC. All suggestions were incorporated in the revised version as follows below.

None of this material has been published or is under consideration for publication elsewhere. All the authors are aware of and in agreement with the current instructions and condition of this Journal.

In the present article we report on the effect of butoconazole in experimental infection by Trypanosoma cruzi in vitro and in vivo.

The treatment of Chagas disease is composed mainly by old and toxic drugs in addition to the occurrence of naturally nitroderivative resistant parasites highlighting the urgent need to find more effective and safer therapies. In this context, the search for new drugs for this neglected tropical disease is urgently needed and we think that our contribution merits publication in MEIOC.

We are thankful for the reviewers' comments, which have certainly contributed to improving the quality of the manuscript.

With best regards,

Yours sincerely,

Dr. Maria de Nazaré C. Soeiro

Laboratório de Biologia Celular

Instituto Oswaldo Cruz, Fundação Oswaldo Cruz

Av. Brasil 4365 - CEP 21040-360

Rio de Janeiro - Brasil

Email: soeiro@ioc.fiocruz.br

Tel. +55 21 25621368

Rebuttal to referee's comments

REVIEWER COMMENTS:

Reviewer: 1

This manuscript presents interesting and relevant results in the field, as it is a comprehensive study that evaluates the in vitro and in vivo effects of the CYP51 inhibitor BTZ against T. cruzi, in addition to its pharmacokinetic profile. The study is conducted with a particular focus on the efficacy of BTZ when used as a monotherapy or in combination with other agents. While the results were not as promising as one might have hoped, I still believe that the work has merit for publication in Memórias do Instituto Oswaldo Cruz. Nevertheless, certain aspects of the study require further attention before it can be accepted for publication. These are outlined below.

Answer: We thank the Referee's comments and included all suggested aspects as follows:

1. The term "more active" should be replaced with "more potent." For example, the statement "23-fold more active than BZ" in lines 39 and 40 requires clarification. A review of all similar cases within the manuscript is required.

Answer: We agree. Done.

2. Line 70: The term "anti-tripanosomicidal" is erroneous. The suffix "cidal" denotes the act of causing death; thus, the prefix "anti-" is not applicable (e.g., bactericidal vs. anti-bactericidal). The correct form is "trypanosomicidal" or "trypanocidal" (the latter is preferable).

Answer: We fully agree. Only "trypanosomicidal" has been retained in the manuscript. Thank you for the observation.

3. The rationale behind the administration of BTZ via the IP route is unclear, given that the oral route is typically preferred for the development of antichagasic drugs. It is understood that the oral bioavailability of the compound in question is likely to be low due to its low aqueous solubility. However, this information should be presented in the article for the benefit of the reader.

Answer: Thank you for your suggestion. We have made explicit the reason for the selection of the i.p. route in the methodological section of the revised manuscript.

4. In lines 241 to 243, it is stated that "A fresh solution of BTZ 100 mg/kg diluted in DMSO (2mL/kg, BIOPACK) was administered intraperitoneally (i.p.)." However, the mg/kg unit is used for dose, not for concentration in solution. It appears that there has been a misunderstanding between the units used to express concentration and dose. This should be corrected.

Answer: Thank you for this relevant observation.. The sentence was revised as follows: "A fresh solution of BTZ at 50 mg/mL in DMSO (BIOPACK) was administered intraperitoneally (i.p.) at an injection volume of 2 mL/kg, achieving a final dose of 100 mg/kg."

5. It would be beneficial to present the in vitro inhibition curve for BTZ, particularly to ascertain the extent of its maximum inhibition. CYP51 inhibitors typically exhibit a greater trypanostatic than trypanocidal effect, which may contribute to the observed lack of efficacy of BTZ in this study, particularly when considering the Cmax observed in the PK study.

Answer: We agree. The curve has been Included in the revised version.

6. Line 335: It is stated that the results are in Table I, yet this is not included in the file that was sent. In fact, none of the tables that were referenced are present; only the figures are presented at the end of the manuscript.

Answer: I am sorry for any upload problem. The tables have now included withiin the main manuscript file

Reviewer: 2

a) Adequacy of the abstract;

Adequate. However, significant improvements must be made to the results to consider them adequate for publication.

b) Originality and importance of the contribution for the development of the field of study

The article's proposition will contribute to the field of neglected diseases drug discovery since the authors performed pharmacokinetics analysis - which still is overlooked by most of the academic research groups - in addition to in vitro potency with very good methods and carry out interesting in vivo assays. Nonetheless, the authors could present a better manuscript by taking more advantage of the several techniques used and presenting poor data or not presented at all, considering that five of nine results are tables not provided by the authors.

Answer: We thank the Referee's comments and included all suggested aspects as follows:

c) Methodology, results and discussion;

All results the authors intend to show as tables must be demonstrated as graphs because tables are unreliable for verifying the quality and robustness of the intended results. Tables could be maintained to summarize the results better.

Answer: We agree. Figures were included in the revised MS.

The authors suppress entirely the error values in one of the most important results, the parasitemia graphs. It is known that Pizzi-Brenner quantification in mice models, particularly in non-isogenic lineages such as Swiss mice, generates highly variable data. Therefore, it is only acceptable to show these results with the error values.

Answer: We agree. Erro values were included in the parasitemia graphs of the revised MS.

Considerations about experiment design: The authors use several treatment regimens, leading to several groups with a few mice per group. Thus, the robustness of the data was decreased, especially considering the high intrinsic variance of the method chosen to access the parasite load. No statistical analysis was provided in the text, or the graphs and statements were made based on weak data.

Answer: Thank you for the comments. In the efficacy experiments: in the first set of assays, all groups were done with 5 animals per group. In the second assay, the number of animals per group was $\geq 3$ and was set as recommend by our institutional ethics committee. Also, all animal groups were evaluated under the same methods to assess parasite load performed by qPCR of mice blood. The statistical analysis was included in the revised MS. p values have been included in the revised version, to made explicit the level of statistical significance of each observation.

The statement content between lines 69 and 71 needs to be clarified.

Separate numbers from units in all text.

Detailed information such as strain, DTU, and abbreviation should be removed from the subtitle and included in the text to clarify the main goal of the assay

Answer: We agree. The statement was revised and clarified. The numbers were separated from units in all text. The detailed information such as strain, DTU, and abbreviation were removed from the subtitle and included in the text.

Despite several assays using different mammalian cell lines and parasite strains, the authors did not show any graph results regarding the cytotoxicity in vitro data.

Provide the cytotoxicity and anti-trypanocide activity data as a fit curve graph. Since GraphPad analysis provides a value of EC50 in almost all cases, even when the curve is not properly fitted with upper and bottom plateus, the EC50 value by itself is unreliable. Considering that the authors indicate in material and methods that the assays were performed at least twice, the EC50 must demonstrate the SD between those assays.

Answer: The SD has been provided in all the tables. Also, the figures regarding the cytotoxicity and anti-trypanocide activity data as a fit curve graph are now included as suggested.

To ensure a compatible comparison of data obtained in this study with others in the literature, please inform EC50 obtained for BZ in all assays in the text, as much as it was informed for the test compound.

Answer: The EC50 values of BZ were added as well as all SD.

To make it easier and straightforward, the comparison with other parameters used in drug discovery and data integration by software used in the field of medicinal chemistry has used EC50 values as pEC50 (pEC50 = –Log (EC50 [M])). Therefore, include the pEC50 values in the tables along with EC50.

Answer: The pEC50 values were added to the tables.

Regarding the anti-parasitic assays, the range used to test the parasites with BZ (as control) from 0 to 10uM is unlikely to reach the inhibition plateau needed to properly calculate EC50 since 100% (or nearly) was probably achieved. Please provide the graphs demonstrating efficacy concentrations.

Answer: The graphics are now included.

Please, provide complete information about the source of bloodstream trypomastigotes in the section about in vitro bloodstream trypomastigotes assay, including host species, initial inoculum, day of infection, method of blood puncture, anesthesia method, and ethics committee protocol approval number.

Answer: All the requested information was included.

Authors mention in line 209 seven dilutions, including one of 1:2; however, in line 204/205, six dilutions were mentioned, and none of them are 1:2 factor. Please rewrite the paragraph between lines 202 and 213 to improve reader comprehension. Line 204/205: Include the concentration values in uM since, traditionally, the isobologram is a graph of concentrations.

Answer: In the drug combination assay, different BTZ and BZ ratios (5:0; 4+1; 3+2; 2+3; 1+4 and 0+5) were done and then a serial dilution (1:2) was performed. To clarify this point, the sentence was rewritten in the revised version.

Line 275: Specify the value of inoculum to avoid any misled interpretation. (ie: 1x10^4 or 5x10^4)

Answer: Done.

Can the authors explain the chosen suboptimal concentrations of BZ for combo treatment? BZ 10 mg/kg/ for 5 days in the acute stage was already proven to have little contribution to treatment outcome due to extremely low blood exposure.

Answer: A ten-fold lower optimal dose of BZ was chosen as it only mildly protects against T.cruzi experimental infection, allowing a more clearly observation of any benefit of its association with BTZ. The information was included.

To improve animal well-being, mortality rate assays are strongly discouraged from being performed without humanized end points guided by clinical and behavioral assessments such as body temperature, lack of movement rates, and grimace scale to measure rodent pain. Did the authors apply any of these measures to secure experiment refinement?

Answer: All methodology (including mortality rates) was performed according to the Ethical approval by Fiocruz committee (License CEUA L038-2017).

Until line 299, no reagent supplier was informed, except by tested compounds. Please provide all suppliers for all commercial reagents used to ensure text uniformity and experimental reproducibility. Place of origin is not necessary.

Answer: Done.

Please move the ethics statement before the description of the in vivo methods.

Answer: Done.

Lines 354 to 360 belong to the description of the methods and should not be placed in the results section.

Answer: Done.

The authors cannot state that the death of BTZ is due to parasitic infection since they did not include a group treated and non-infected, and no information about toxicity was provided. If it is not possible to perform the experiment including this group, data from the literature is useful since it is a drug already approved. However, authors must provide information with similar models and treatment regimens regarding mice strain, age, BTZ dose, and route.

Answer: We agree. Although BTZ is only used topically as antifungal with mild or none side effects (Fromtling RA. Overview of medically important antifungal azole derivatives. Clin Microbiol Rev. 1988 Apr;1(2):187-217. doi: 10.1128/CMR.1.2.187. PMID: 3069196; PMCID: PMC3580420, as no uninfected and BTZ group was evaluated, the sentence was deleted.

Refine the text to improve readers' comprehension. Describe the treatment regimen instead of more generic terms. ie: BZ optimal dose for BZ 100 mg/kg.

Answer: Following your advice, the description was revised aiming to improve readers' comprehension.

qPCR results: Improve text consistency, providing the results as positive/total or negative/total. ie: 3/3, 0/4. Since qPCR is a quantitative technique, the authors can explain the reason why they chose to show the data as a table instead of plotting them as graphs exhibiting the quantified amount of DNA of each group.

Answer: all data from qPCR were related to the number of positive/total animals with corresponding to the Eq.Par/mg values. To be clearer, graphs replaced the Table as recommended.

d) References;

Please revise the references addressing the statements in the article or websites that the main contributions are the subject of the information stated in the manuscript. For instance, but not restricted to these examples:

The reference of the DNDi website indeed contains the referred information; however, studies that properly assess this knowledge should be referred to instead, along with the factsheets of health and research organizations.

References 4,5, and 6 contain information about Chagas disease prevalence. However, the main subjects covered by references 4 and 5 encompass other knowledge areas, and the referred statements are only part of the introduction.

Revise all references in sequences since they are presented in different styles—for instance, line 102 and line 106.

Line 107 and 108 - Instead of citing more wide reviews as references 14, 25, 26, 27, 28, use other articles that discuss more deeply and are the original source of scientific knowledge about CYP51 inhibitors as treatment of CD. Including them in the introduction can bring a more refined discussion. This action intends to address the proper researcher recognition and avoid the propagation of misleading data interpretation.

Citation error: Revise the reference number 16 because the statement between lines 133 and 135 does not match the article content. The reference the author probably wanted to cite is number 18 instead of No 16, and this error was noticed more than once. Please revise all citation orders; since other misled occurrences were noticed, such as reference number 31; from this point forward, it is not possible to verify the reference information.

Answer: All references were revised takin into consideration the reviewer's comments. Thank you for your advice.

e) Figures and tables.

Figure 2: Why did the authors plot the isobologram with segmenting for both axes since the concentration is linear? The data of 5:0 and 5:0 is not plotted in the graph. Show axis titles with complete descriptions of units.

Answer: The graph was revised.

Figure 3: Provide only one graph with the total and unbound fraction data. Include in the graph as 'cutoff' the EC50 value. Despite this, it is important to consider that the EC50 alone is not a reliable value since it should be considered the protein binding capacity in the culture medium.

Answer: Thank you for the suggestion. Figures 3A and 3B have now been integrated into a single graph, following the reviewer's comment. The EC50 has been also included. We agree with you that the figure is more informative in this new version. We also completely agree with your comment on the importance the drug protein binding in the culture medium. In any case, the "unbound EC50" would be below the estimated EC50, which considers both bound and unbound drug in the medium. We have included a sentence discussing this in lines 340-343 of the revised manuscript.

Figure 4: There are no error bars in any parasitemia data, which is impossible since several mice were used in this experiment. Statistical analysis is not shown or no difference was observed in the data.

Answer: Done. Statistical analysis was included in the revised MS.

The reason for the separated graphs in A and C needs to be clarified.

In the axis title, use complete and precise words. Use "trypomastigotes" instead of "par" Include the treatment period in the graphs as a shaded area or dotted lines to represent the treatment.

Answer: The graphs are separated because they represent different assays (1 and 2) as now indicated in the legend. Par replaced by trypomastigotes. Also, treatment period was included.

Revise the legend's information carefully, aiming to be precise and more informative. It presents errors in the inoculum, and non-informative/ redundant statements were made.

Answer: All legends were revised.

## SECOND REVIEW ROUND

REVIEWERS' COMMENTS

**REVIEWER #1**

Reviewer comments: The changes in text and figures improved considerably in the manuscript. Nevertheless, significant concerns regarding EC50 values remain. In this new version, the authors provide the dose-response curves. Unlike BZ, the BTZ does not exhibit evenly distributed points to secure a trustful value of EC50.

Considering that the EC50 is an important parameter for other experiments in this manuscript and that other researchers will base their experiments on this value in the future, it is important to provide more consistent data.

The second point in the BTZ curve achieved a value near the plate. One possible explanation is the low solubility of the compound. In order to overcome this issue or any other possible, I request to perform this experiment again, taking in consideration to find intermediate values between 0 and 80% parasite death for BTZ.

Careful revising Figures, Legends and Results: The text's explanation of figures and legends does not match. In lines 314 to 316, the authors describe that Figure 2 is the "cytotoxic profile of BTZ on L929 cells", implying an assay in non-infected cells. However, the legend in Fig 2 says: "against intracellular amastigotes of T. cruzi".

In lines 316 to 320, the potency results against parasites are presented as illustrated in Fig 3. Nonetheless, the legend describes "L929 mammalian cell lines". Considering the concentration values in the graph, it is possible to realize that the legends are misplaced. Please revise all information provided as a result from all the figures and tables.

Figure 6 was improved as requested. Please consider the following changes:

The X-axis should be changed to "day post-infection" since the authors refer to infection days as "dpi".

The symbol used to demonstrate the period of drug administration is called "dashed lines"; however, the legend is described as "Dot lines".

Include in the legend the number of mice used in each group.

Revise written English: comma in numbers instead of dots, units, subscripts, superscripts, and other minor issues.

AUTHORS' RESPONSE TO THE REVIEWERS

Dear Editor,

Please, find attached the revised manuscript entitled "In vitro and in vivo studies on the activity and selectivity of butoconazole in experimental infection by Trypanosoma cruzi" by Gabriela Rodrigues Leite, Denise da Gama Jaen Batista, Marcos Meuser Batista, Krislayne Nunes da Costa, Tomás Mac Loughlin, Emilia M. Barrionuevo, Alan Talevi, Lucas N. Aberca, Otacilio C. Moreira, Amanda Faier-Pereira, Beatriz Iandra da Silva Ferreira, and Maria de Nazaré Correia Soeiro to be considered for publication in MEIOC. All suggestions were incorporated in the revised version as follows below.

None of this material has been published or is under consideration for publication elsewhere. All the authors are aware of and in agreement with the current instructions and condition of this Journal.

In the present article we report on the effect of butoconazole in experimental infection by Trypanosoma cruzi in vitro and in vivo.

The treatment of Chagas disease is composed mainly by old and toxic drugs in addition to the occurrence of naturally nitroderivative resistant parasites highlighting the urgent need to find more effective and safer therapies. In this context, the search for new drugs for this neglected tropical disease is urgently needed and we think that our contribution merits publication in MEIOC.

We are thankful for the reviewers' comments, which have certainly contributed to improving the quality of the manuscript.

With best regards,

Yours sincerely,

Dr. Maria de Nazaré C. Soeiro

Laboratório de Biologia Celular

Instituto Oswaldo Cruz, Fundação Oswaldo Cruz

Av. Brasil 4365 - CEP 21040-360

Rio de Janeiro - Brasil

Email: soeiro@ioc.fiocruz.br

Tel. +55 21 25621368

Rebuttal to referee's comments
REVIEWER COMMENTS:
Reviewer: 1

The changes in text and figures improved considerably in the manuscript. Nevertheless, significant concerns regarding EC50 values remain. In this new version, the authors provide the dose-response curves. Unlike BZ, the BTZ does not exhibit evenly distributed points to secure a trustful value of EC50. Considering that the EC50 is an important parameter for other experiments in this manuscript and that other researchers will base their experiments on this value in the future, it is important to provide more consistent data. The second point in the BTZ curve achieved a value near the plate. One possible explanation is the low solubility of the compound. In order to overcome this issue or any other possible, I request to perform this experiment again, taking in consideration to find intermediate values between 0 and 80% parasite death for BTZ.

Answer: Thanks for the comments and contribution. New assays were performed and the data included.

Careful revising Figures, Legends and Results: The text's explanation of figures and legends does not match. In lines 314 to 316, the authors describe that Figure 2 is the "cytotoxic profile of BTZ on L929 cells", implying an assay in non-infected cells. However, the legend in Fig 2 says: "against intracellular amastigotes of T. cruzi".

Answer: Thanks. The text was revised.

In lines 316 to 320, the potency results against parasites are presented as illustrated in Fig 3. Nonetheless, the legend describes "L929 mammalian cell lines". Considering the concentration values in the graph, it is possible to realize that the legends are misplaced. Please revise all information provided as a result from all the figures and tables.

Answer: Thanks. The text was revised.

Figure 6 was improved as requested. Please consider the following changes:

The X-axis should be changed to "day post-infection" since the authors refer to infection days as "dpi". The symbol used to demonstrate the period of drug administration is called "dashed lines"; however, the legend is described as "Dot lines".

Include in the legend the number of mice used in each group.

Answer: Thanks. The legend and figure 6 were revised.

Revise written English: comma in numbers instead of dots, units, subscripts, superscripts, and other minor issues.

Answer: Thanks. The text was revised.

## THIRD REVIEW ROUND

REVIEWERS' COMMENTS

### REVIEWER #1

Reviewer comments: The manuscript is able to be accepted in its current form.

### REVIEWER #2

Reviewer comments: The addition of a new experiment and the modifications made in the text were appreciated. However, it was noticed that changes in the overall values or even changes in the standard deviation of IC50 in the text or tables were not performed.

Congratulations on the work, and I hope that the considerations made improve the scientific knowledge of the research group.

