## [Reviewer Report · FIRST REVIEW ROUND - REVIEWERS COMMENTS]

## REVIEWER #1

This manuscript presents interesting and relevant results in the field, as it is a comprehensive study that evaluates the in vitro and in vivo effects of the CYP51 inhibitor BTZ against T. cruzi, in addition to its pharmacokinetic profile. The study is conducted with a particular focus on the efficacy of BTZ when used as a monotherapy or in combination with other agents. While the results were not as promising as one might have hoped, I still believe that the work has merit for publication in Memórias do Instituto Oswaldo Cruz. Nevertheless, certain aspects of the study require further attention before it can be accepted for publication. These are outlined below.

The term “more active” should be replaced with “more potent.” For example, the statement “23-fold more active than BZ” in lines 39 and 40 requires clarification. A review of all similar cases within the manuscript is required.

Line 70: The term “anti-tripanosomicidal” is erroneous. The suffix “cidal” denotes the act of causing death; thus, the prefix “anti-” is not applicable (e.g., bactericidal vs. anti-bactericidal). The correct form is “trypanosomicidal” or “trypanocidal” (the latter is preferable).

The rationale behind the administration of BTZ via the IP route is unclear, given that the oral route is typically preferred for the development of antichagasic drugs. It is understood that the oral bioavailability of the compound in question is likely to be low due to its low aqueous solubility. However, this information should be presented in the article for the benefit of the reader.

In lines 241 to 243, it is stated that “A fresh solution of BTZ 100 mg/kg diluted in DMSO (2mL/kg, BIOPACK) was administered intraperitoneally (i.p.).” However, the mg/kg unit is used for dose, not for concentration in solution. It appears that there has been a misunderstanding between the units used to express concentration and dose. This should be corrected.

It would be beneficial to present the in vitro inhibition curve for BTZ, particularly to ascertain the extent of its maximum inhibition. CYP51 inhibitors typically exhibit a greater trypanostatic than trypanocidal effect, which may contribute to the observed lack of efficacy of BTZ in this study, particularly when considering the Cmax observed in the PK study.

Line 335: It is stated that the results are in Table I, yet this is not included in the file that was sent. In fact, none of the tables that were referenced are present; only the figures are presented at the end of the manuscript.

## REVIEWER #2

a) Adequacy of the abstract;

Adequate. However, significant improvements must be made to the results to consider them adequate for publication.

b) Originality and importance of the contribution for the development of the field of study

The article’s proposition will contribute to the field of neglected diseases drug discovery since the authors performed pharmacokinetics analysis - which still is overlooked by most of the academic research groups - in addition to in vitro potency with very good methods and carry out interesting in vivo assays. Nonetheless, the authors could present a better manuscript by taking more advantage of the several techniques used and presenting poor data or not presented at all, considering that five of nine results are tables not provided by the authors.

c) Methodology, results and discussion;

All results the authors intend to show as tables must be demonstrated as graphs because tables are unreliable for verifying the quality and robustness of the intended results. Tables could be maintained to summarize the results better.

The authors suppress entirely the error values in one of the most important results, the parasitemia graphs. It is known that Pizzi-Brenner quantification in mice models, particularly in non-isogenic lineages such as Swiss mice, generates highly variable data. Therefore, it is only acceptable to show these results with the error values.

Considerations about experiment design: The authors use several treatment regimens, leading to several groups with a few mice per group. Thus, the robustness of the data was decreased, especially considering the high intrinsic variance of the method chosen to access the parasite load. No statistical analysis was provided in the text, or the graphs and statements were made based on weak data.

The statement content between lines 69 and 71 needs to be clarified.

Separate numbers from units in all text.

Detailed information such as strain, DTU, and abbreviation should be removed from the subtitle and included in the text to clarify the main goal of the assay

Despite several assays using different mammalian cell lines and parasite strains, the authors did not show any graph results regarding the cytotoxicity in vitro data.

Provide the cytotoxicity and anti-trypanocide activity data as a fit curve graph. Since GraphPad analysis provides a value of EC50 in almost all cases, even when the curve is not properly fitted with upper and bottom plateus, the EC50 value by itself is unreliable. Considering that the authors indicate in material and methods that the assays were performed at least twice, the EC50 must demonstrate the SD between those assays.

To ensure a compatible comparison of data obtained in this study with others in the literature, please inform EC50 obtained for BZ in all assays in the text, as much as it was informed for the test compound.

To make it easier and straightforward, the comparison with other parameters used in drug discovery and data integration by software used in the field of medicinal chemistry has used EC50 values as pEC50 (pEC50 = –Log (EC50 [M])). Therefore, include the pEC50 values in the tables along with EC50.

Regarding the anti-parasitic assays, the range used to test the parasites with BZ (as control) from 0 to 10uM is unlikely to reach the inhibition plateau needed to properly calculate EC50 since 100% (or nearly) was probably achieved. Please provide the graphs demonstrating efficacy concentrations. Two articles that can instruct the authors to provide this graph according to pharmacology guidelines:

Neubig RR, Spedding M, Kenakin T, Christopoulos A; International Union of Pharmacology Committee on Receptor Nomenclature and Drug Classification. International Union of Pharmacology Committee on Receptor Nomenclature and Drug Classification. XXXVIII. Update on terms and symbols in quantitative pharmacology. Pharmacol Rev. 2003 Dec;55(4):597-606. doi: 10.1124/pr.55.4.4. PMID: 14657418.

Sebaugh JL. Guidelines for accurate EC50/IC50 estimation. Pharm Stat. 2011 Mar-Apr;10(2):128-34. doi: 10.1002/pst.426. PMID: 22328315.

Please, provide complete information about the source of bloodstream trypomastigotes in the section about in vitro bloodstream trypomastigotes assay, including host species, initial inoculum, day of infection, method of blood puncture, anesthesia method, and ethics committee protocol approval number.

Authors mention in line 209 seven dilutions, including one of 1:2; however, in line 204/205, six dilutions were mentioned, and none of them are 1:2 factor. Please rewrite the paragraph between lines 202 and 213 to improve reader comprehension.

Line 204/205: Include the concentration values in uM since, traditionally, the isobologram is a graph of concentrations.

Line 275: Specify the value of inoculum to avoid any misled interpretation. (ie: $1\times 10^4$ or $5\times 10^4$)

Can the authors explain the chosen suboptimal concentrations of BZ for combo treatment? BZ 10 mg/kg/ for 5 days in the acute stage was already proven to have little contribution to treatment outcome due to extremely low blood exposure.

To improve animal well-being, mortality rate assays are strongly discouraged from being performed without humanized end points guided by clinical and behavioral assessments such as body temperature, lack of movement rates, and grimace scale to measure rodent pain. Did the authors apply any of these measures to secure experiment refinement?

Until line 299, no reagent supplier was informed, except by tested compounds. Please provide all suppliers for all commercial reagents used to ensure text uniformity and experimental reproducibility. Place of origin is not necessary.

Please move the ethics statement before the description of the in vivo methods.

Lines 354 to 360 belong to the description of the methods and should not be placed in the results section.

The authors cannot state that the death of BTZ is due to parasitic infection since they did not include a group treated and non-infected, and no information about toxicity was provided. If it is not possible to perform the experiment including this group, data from the literature is useful since it is a drug already approved. However, authors must provide information with similar models and treatment regimens regarding mice strain, age, BTZ dose, and route.

Refine the text to improve readers’ comprehension. Describe the treatment regimen instead of more generic terms. ie: BZ optimal dose for BZ 100 mg/kg.

qPCR results: Improve text consistency, providing the results as positive/total or negative/total. ie: 3/3, 0/4.

Since qPCR is a quantitative technique, the authors can explain the reason why they chose to show the data as a table instead of plotting them as graphs exhibiting the quantified amount of DNA of each group.

d) References;

Please revise the references addressing the statements in the article or websites that the main contributions are the subject of the information stated in the manuscript. For instance, but not restricted to these examples:

The reference of the DNDi website indeed contains the referred information; however, studies that properly assess this knowledge should be referred to instead, along with the factsheets of health and research organizations.

References 4,5, and 6 contain information about Chagas disease prevalence. However, the main subjects covered by references 4 and 5 encompass other knowledge areas, and the referred statements are only part of the introduction.

Revise all references in sequences since they are presented in different styles—for instance, line 102 and line 106.

Line 107 and 108 - Instead of citing more wide reviews as references 14, 25, 26, 27, 28, use other articles that discuss more deeply and are the original source of scientific knowledge about CYP51 inhibitors as treatment of CD. Including them in the introduction can bring a more refined discussion. This action intends to address the proper researcher recognition and avoid the propagation of misleading data interpretation.

Citation error: Revise the reference number 16 because the statement between lines 133 and 135 does not match the article content. The reference the author probably wanted to cite is number 18 instead of No 16, and this error was noticed more than once. Please revise all citation orders; since other misled occurrences were noticed, such as reference number 31; from this point forward, it is not possible to verify the reference information.

e) Figures and tables.

Figure 2: Why did the authors plot the isobologram with segmenting for both axes since the concentration is linear?

The data of 5:0 and 5:0 is not plotted in the graph.

Show axis titles with complete descriptions of units.

Figure 3: Provide only one graph with the total and unbound fraction data. Include in the graph as ‘cutoff’ the EC50 value. Despite this, it is important to consider that the EC50 alone is not a reliable value since it should be considered the protein binding capacity in the culture medium.

Figure 4: There are no error bars in any parasitemia data, which is impossible since several mice were used in this experiment. Statistical analysis is not shown or no difference was observed in the data.

The reason for the separated graphs in A and C needs to be clarified.

In the axis title, use complete and precise words. Use “trypomastigotes” instead of “par”

Include the treatment period in the graphs as a shaded area or dotted lines to represent the treatment.

Revise the legend’s information carefully, aiming to be precise and more informative. It presents errors in the inoculum, and non-informative/ redundant statements were made.

Please see the pattern of legends used for research groups experienced in Chagas disease proof-of-concept assays.

Francisco AF, Jayawardhana S, Lewis MD, White KL, Shackleford DM, Chen G, Saunders J, Osuna-Cabello M, Read KD, Charman SA, Chatelain E, Kelly JM. Nitroheterocyclic drugs cure experimental Trypanosoma cruzi infections more effectively in the chronic stage than in the acute stage. Sci Rep. 2016 Oct 17;6:35351. doi: 10.1038/srep35351. PMID: 27748443; PMCID: PMC5066210.

---

## [Author Response · AUTHORS RESPONSE TO REVIEWERS]

## Dear Editor,

Please, find attached the revised manuscript entitled “In vitro and in vivo studies on the activity and selectivity of butoconazole in experimental infection by Trypanosoma cruzi” by Gabriela Rodrigues Leite, Denise da Gama Jaen Batista, Marcos Meuser Batista, Krislayne Nunes da Costa, Tomás Mac Loughlin, Emilia M. Barrionuevo, Alan Talevi, Lucas N. Aberca, Otacilio C. Moreira, Amanda Faier-Pereira, Beatriz Iandra da Silva Ferreira, and Maria de Nazaré Correia Soeiro to be considered for publication in MEIOC. All suggestions were incorporated in the revised version as follows below.

None of this material has been published or is under consideration for publication elsewhere. All the authors are aware of and in agreement with the current instructions and condition of this Journal.

In the present article we report on the effect of butoconazole in experimental infection by Trypanosoma cruzi in vitro and in vivo.

The treatment of Chagas disease is composed mainly by old and toxic drugs in addition to the occurrence of naturally nitroderivative resistant parasites highlighting the urgent need to find more effective and safer therapies. In this context, the search for new drugs for this neglected tropical disease is urgently needed and we think that our contribution merits publication in MEIOC.

We are thankful for the reviewers’ comments, which have certainly contributed to improving the quality of the manuscript.

With best regards,

Yours sincerely,

Dr. Maria de Nazaré C. Soeiro

---

## [Reviewer Report · REVIEWERS COMMENTS]

## Reviewer: 1

This manuscript presents interesting and relevant results in the field, as it is a comprehensive study that evaluates the in vitro and in vivo effects of the CYP51 inhibitor BTZ against T. cruzi, in addition to its pharmacokinetic profile. The study is conducted with a particular focus on the efficacy of BTZ when used as a monotherapy or in combination with other agents. While the results were not as promising as one might have hoped, I still believe that the work has merit for publication in Memórias do Instituto Oswaldo Cruz. Nevertheless, certain aspects of the study require further attention before it can be accepted for publication. These are outlined below.

Answer: We thank the Referee’s comments and included all suggested aspects as follows:

1. The term “more active” should be replaced with “more potent.” For example, the statement “23-fold more active than BZ” in lines 39 and 40 requires clarification. A review of all similar cases within the manuscript is required.

Answer: We agree. Done.

2. Line 70: The term “anti-tripanosomicidal” is erroneous. The suffix “cidal” denotes the act of causing death; thus, the prefix “anti-” is not applicable (e.g., bactericidal vs. anti-bactericidal). The correct form is “trypanosomicidal” or “trypanocidal” (the latter is preferable).

Answer: We fully agree. Only “trypanosomicidal” has been retained in the manuscript. Thank you for the observation.

3. The rationale behind the administration of BTZ via the IP route is unclear, given that the oral route is typically preferred for the development of antichagasic drugs. It is understood that the oral bioavailability of the compound in question is likely to be low due to its low aqueous solubility. However, this information should be presented in the article for the benefit of the reader.

Answer: Thank you for your suggestion. We have made explicit the reason for the selection of the i.p. route in the methodological section of the revised manuscript.

4. In lines 241 to 243, it is stated that “A fresh solution of BTZ 100 mg/kg diluted in DMSO (2mL/kg, BIOPACK) was administered intraperitoneally (i.p.).” However, the mg/kg unit is used for dose, not for concentration in solution. It appears that there has been a misunderstanding between the units used to express concentration and dose. This should be corrected.

Answer: Thank you for this relevant observation.. The sentence was revised as follows: “A fresh solution of BTZ at 50 mg/mL in DMSO (BIOPACK) was administered intraperitoneally (i.p.) at an injection volume of 2 mL/kg, achieving a final dose of 100 mg/kg.”

5. It would be beneficial to present the in vitro inhibition curve for BTZ, particularly to ascertain the extent of its maximum inhibition. CYP51 inhibitors typically exhibit a greater trypanostatic than trypanocidal effect, which may contribute to the observed lack of efficacy of BTZ in this study, particularly when considering the Cmax observed in the PK study.

Answer: We agree. The curve has been Included in the revised version.

6. Line 335: It is stated that the results are in Table I, yet this is not included in the file that was sent. In fact, none of the tables that were referenced are present; only the figures are presented at the end of the manuscript.

Answer: I am sorry for any upload problem. The tables have now included withiin the main manuscript file

## Reviewer: 2

a) Adequacy of the abstract;

Adequate. However, significant improvements must be made to the results to consider them adequate for publication.

b) Originality and importance of the contribution for the development of the field of study

The article’s proposition will contribute to the field of neglected diseases drug discovery since the authors performed pharmacokinetics analysis - which still is overlooked by most of the academic research groups - in addition to in vitro potency with very good methods and carry out interesting in vivo assays. Nonetheless, the authors could present a better manuscript by taking more advantage of the several techniques used and presenting poor data or not presented at all, considering that five of nine results are tables not provided by the authors.

Answer: We thank the Referee’s comments and included all suggested aspects as follows:

c) Methodology, results and discussion;

All results the authors intend to show as tables must be demonstrated as graphs because tables are unreliable for verifying the quality and robustness of the intended results. Tables could be maintained to summarize the results better.

Answer: We agree. Figures were included in the revised MS.

The authors suppress entirely the error values in one of the most important results, the parasitemia graphs. It is known that Pizzi-Brenner quantification in mice models, particularly in non-isogenic lineages such as Swiss mice, generates highly variable data. Therefore, it is only acceptable to show these results with the error values.

Answer: We agree. Erro values were included in the parasitemia graphs of the revised MS.

Considerations about experiment design: The authors use several treatment regimens, leading to several groups with a few mice per group. Thus, the robustness of the data was decreased, especially considering the high intrinsic variance of the method chosen to access the parasite load. No statistical analysis was provided in the text, or the graphs and statements were made based on weak data.

Answer: Thank you for the comments. In the efficacy experiments: in the first set of assays, all groups were done with 5 animals per group. In the second assay, the number of animals per group was $\ge 3$ and was set as recommend by our institutional ethics committee. Also, all animal groups were evaluated under the same methods to assess parasite load performed by qPCR of mice blood. The statistical analysis was included in the revised MS. p values have been included in the revised version, to made explicit the level of statistical significance of each observation.

The statement content between lines 69 and 71 needs to be clarified.

Separate numbers from units in all text.

Detailed information such as strain, DTU, and abbreviation should be removed from the subtitle and included in the text to clarify the main goal of the assay

Answer: We agree. The statement was revised and clarified. The numbers were separated from units in all text. The detailed information such as strain, DTU, and abbreviation were removed from the subtitle and included in the text.

Despite several assays using different mammalian cell lines and parasite strains, the authors did not show any graph results regarding the cytotoxicity in vitro data.

Provide the cytotoxicity and anti-trypanocide activity data as a fit curve graph. Since GraphPad analysis provides a value of EC50 in almost all cases, even when the curve is not properly fitted with upper and bottom plateus, the EC50 value by itself is unreliable. Considering that the authors indicate in material and methods that the assays were performed at least twice, the EC50 must demonstrate the SD between those assays.

Answer: The SD has been provided in all the tables. Also, the figures regarding the cytotoxicity and anti-trypanocide activity data as a fit curve graph are now included as suggested.

To ensure a compatible comparison of data obtained in this study with others in the literature, please inform EC50 obtained for BZ in all assays in the text, as much as it was informed for the test compound.

Answer: The EC50 values of BZ were added as well as all SD.

To make it easier and straightforward, the comparison with other parameters used in drug discovery and data integration by software used in the field of medicinal chemistry has used EC50 values as pEC50 (pEC50 = –Log (EC50 [M])). Therefore, include the pEC50 values in the tables along with EC50.

Answer: The pEC50 values were added to the tables.

Regarding the anti-parasitic assays, the range used to test the parasites with BZ (as control) from 0 to 10uM is unlikely to reach the inhibition plateau needed to properly calculate EC50 since 100% (or nearly) was probably achieved. Please provide the graphs demonstrating efficacy concentrations.

Answer: The graphics are now included.

Please, provide complete information about the source of bloodstream trypomastigotes in the section about in vitro bloodstream trypomastigotes assay, including host species, initial inoculum, day of infection, method of blood puncture, anesthesia method, and ethics committee protocol approval number.

Answer: All the requested information was included.

Authors mention in line 209 seven dilutions, including one of 1:2; however, in line 204/205, six dilutions were mentioned, and none of them are 1:2 factor. Please rewrite the paragraph between lines 202 and 213 to improve reader comprehension. Line 204/205: Include the concentration values in uM since, traditionally, the isobologram is a graph of concentrations.

Answer: In the drug combination assay, different BTZ and BZ ratios (5:0; 4+1; 3+2; 2+3; 1+4 and 0+5) were done and then a serial dilution (1:2) was performed. To clarify this point, the sentence was rewritten in the revised version.

Line 275: Specify the value of inoculum to avoid any misled interpretation. (ie: $1\times 10^4$ or $5\times 10^4$)

Answer: Done.

Can the authors explain the chosen suboptimal concentrations of BZ for combo treatment? BZ 10 mg/kg/ for 5 days in the acute stage was already proven to have little contribution to treatment outcome due to extremely low blood exposure.

Answer: A ten-fold lower optimal dose of BZ was chosen as it only mildly protects against T.cruzi experimental infection, allowing a more clearly observation of any benefit of its association with BTZ. The information was included.

To improve animal well-being, mortality rate assays are strongly discouraged from being performed without humanized end points guided by clinical and behavioral assessments such as body temperature, lack of movement rates, and grimace scale to measure rodent pain. Did the authors apply any of these measures to secure experiment refinement?

Answer: All methodology (including mortality rates) was performed according to the Ethical approval by Fiocruz committee (License CEUA L038-2017).

Until line 299, no reagent supplier was informed, except by tested compounds. Please provide all suppliers for all commercial reagents used to ensure text uniformity and experimental reproducibility. Place of origin is not necessary.

Answer: Done.

Please move the ethics statement before the description of the in vivo methods.

Answer: Done.

Lines 354 to 360 belong to the description of the methods and should not be placed in the results section.

Answer: Done.

The authors cannot state that the death of BTZ is due to parasitic infection since they did not include a group treated and non-infected, and no information about toxicity was provided. If it is not possible to perform the experiment including this group, data from the literature is useful since it is a drug already approved. However, authors must provide information with similar models and treatment regimens regarding mice strain, age, BTZ dose, and route.

Answer: We agree. Although BTZ is only used topically as antifungal with mild or none side effects (Fromtling RA. Overview of medically important antifungal azole derivatives. Clin Microbiol Rev. 1988 Apr;1(2):187-217. doi: 10.1128/CMR.1.2.187. PMID: 3069196; PMCID: PMC3580420, as no uninfected and BTZ group was evaluated, the sentence was deleted.

Refine the text to improve readers’ comprehension. Describe the treatment regimen instead of more generic terms. ie: BZ optimal dose for BZ 100 mg/kg.

Answer: Following your advice, the description was revised aiming to improve readers’ comprehension.

qPCR results: Improve text consistency, providing the results as positive/total or negative/total. ie: 3/3, 0/4. Since qPCR is a quantitative technique, the authors can explain the reason why they chose to show the data as a table instead of plotting them as graphs exhibiting the quantified amount of DNA of each group.

Answer: all data from qPCR were related to the number of positive/total animals with corresponding to the Eq.Par/mg values. To be clearer, graphs replaced the Table as recommended.

d) References;

Please revise the references addressing the statements in the article or websites that the main contributions are the subject of the information stated in the manuscript. For instance, but not restricted to these examples:

The reference of the DNDi website indeed contains the referred information; however, studies that properly assess this knowledge should be referred to instead, along with the factsheets of health and research organizations.

References 4,5, and 6 contain information about Chagas disease prevalence. However, the main subjects covered by references 4 and 5 encompass other knowledge areas, and the referred statements are only part of the introduction.

Revise all references in sequences since they are presented in different styles—for instance, line 102 and line 106.

Line 107 and 108 - Instead of citing more wide reviews as references 14, 25, 26, 27, 28, use other articles that discuss more deeply and are the original source of scientific knowledge about CYP51 inhibitors as treatment of CD. Including them in the introduction can bring a more refined discussion.

This action intends to address the proper researcher recognition and avoid the propagation of misleading data interpretation.

Citation error: Revise the reference number 16 because the statement between lines 133 and 135 does not match the article content. The reference the author probably wanted to cite is number 18 instead of No 16, and this error was noticed more than once. Please revise all citation orders; since other misled occurrences were noticed, such as reference number 31; from this point forward, it is not possible to verify the reference information.

Answer: All references were revised takin into consideration the reviewer’s comments. Thank you for your advice.

e) Figures and tables.

Figure 2: Why did the authors plot the isobologram with segmenting for both axes since the concentration is linear? The data of 5:0 and 5:0 is not plotted in the graph. Show axis titles with complete descriptions of units.

Answer: The graph was revised.

Figure 3: Provide only one graph with the total and unbound fraction data. Include in the graph as ‘cutoff’ the EC50 value. Despite this, it is important to consider that the EC50 alone is not a reliable value since it should be considered the protein binding capacity in the culture medium.

Answer: Thank you for the suggestion. Figures 3A and 3B have now been integrated into a single graph, following the reviewer’s comment. The EC50 has been also included. We agree with you that the figure is more informative in this new version. We also completely agree with your comment on the importance the drug protein binding in the culture medium. In any case, the “unbound EC50” would be below the estimated EC50, which considers both bound and unbound drug in the medium. We have included a sentence discussing this in lines 340-343 of the revised manuscript.

Figure 4: There are no error bars in any parasitemia data, which is impossible since several mice were used in this experiment. Statistical analysis is not shown or no difference was observed in the data.

Answer: Done. Statistical analysis was included in the revised MS.

The reason for the separated graphs in A and C needs to be clarified. In the axis title, use complete and precise words. Use “trypomastigotes” instead of “par” Include the treatment period in the graphs as a shaded area or dotted lines to represent the treatment.

Answer: The graphs are separated because they represent different assays (1 and 2) as now indicated in the legend. Par replaced by trypomastigotes. Also, treatment period was included.

Revise the legend’s information carefully, aiming to be precise and more informative. It presents errors in the inoculum, and non-informative/ redundant statements were made.

Answer: All legends were revised.

---

## [Reviewer Report · REVIEWERS’ COMMENTS]

## REVIEWER #1

Reviewer comments: The changes in text and figures improved considerably in the manuscript. Nevertheless, significant concerns regarding EC50 values remain. In this new version, the authors provide the dose-response curves. Unlike BZ, the BTZ does not exhibit evenly distributed points to secure a trustful value of EC50.

Considering that the EC50 is an important parameter for other experiments in this manuscript and that other researchers will base their experiments on this value in the future, it is important to provide more consistent data.

The second point in the BTZ curve achieved a value near the plate. One possible explanation is the low solubility of the compound. In order to overcome this issue or any other possible, I request to perform this experiment again, taking in consideration to find intermediate values between 0 and 80% parasite death for BTZ.

Careful revising Figures, Legends and Results: The text’s explanation of figures and legends does not match. In lines 314 to 316, the authors describe that Figure 2 is the “cytotoxic profile of BTZ on L929 cells”, implying an assay in non-infected cells. However, the legend in Fig 2 says: “against intracellular amastigotes of T. cruzi”.

In lines 316 to 320, the potency results against parasites are presented as illustrated in Fig 3. Nonetheless, the legend describes “L929 mammalian cell lines”. Considering the concentration values in the graph, it is possible to realize that the legends are misplaced. Please revise all information provided as a result from all the figures and tables.

Figure 6 was improved as requested. Please consider the following changes:

The X-axis should be changed to “day post-infection” since the authors refer to infection days as “dpi”.

The symbol used to demonstrate the period of drug administration is called “dashed lines”; however, the legend is described as “Dot lines”.

Include in the legend the number of mice used in each group.

Revise written English: comma in numbers instead of dots, units, subscripts, superscripts, and other minor issues.

---

## [Author Response · AUTHORS RESPONSE TO REVIEWERS]

## AUTHORS’ RESPONSE TO THE REVIEWERS

Dear Editor,

Please, find attached the revised manuscript entitled “In vitro and in vivo studies on the activity and selectivity of butoconazole in experimental infection by Trypanosoma cruzi” by Gabriela Rodrigues Leite, Denise da Gama Jaen Batista, Marcos Meuser Batista, Krislayne Nunes da Costa, Tomás Mac Loughlin, Emilia M. Barrionuevo, Alan Talevi, Lucas N. Aberca, Otacilio C. Moreira, Amanda Faier-Pereira, Beatriz Iandra da Silva Ferreira, and Maria de Nazaré Correia Soeiro to be considered for publication in MEIOC. All suggestions were incorporated in the revised version as follows below.

None of this material has been published or is under consideration for publication elsewhere. All the authors are aware of and in agreement with the current instructions and condition of this Journal.

In the present article we report on the effect of butoconazole in experimental infection by Trypanosoma cruzi in vitro and in vivo.

The treatment of Chagas disease is composed mainly by old and toxic drugs in addition to the occurrence of naturally nitroderivative resistant parasites highlighting the urgent need to find more effective and safer therapies. In this context, the search for new drugs for this neglected tropical disease is urgently needed and we think that our contribution merits publication in MEIOC.

We are thankful for the reviewers’ comments, which have certainly contributed to improving the quality of the manuscript.

With best regards,

Yours sincerely,

Dr. Maria de Nazaré C. Soeiro

---

## [Reviewer Report · REVIEWERS COMMENTS]

## Reviewer: 1

The changes in text and figures improved considerably in the manuscript. Nevertheless, significant concerns regarding EC50 values remain. In this new version, the authors provide the dose-response curves. Unlike BZ, the BTZ does not exhibit evenly distributed points to secure a trustful value of EC50. Considering that the EC50 is an important parameter for other experiments in this manuscript and that other researchers will base their experiments on this value in the future, it is important to provide more consistent data. The second point in the BTZ curve achieved a value near the plate. One possible explanation is the low solubility of the compound. In order to overcome this issue or any other possible, I request to perform this experiment again, taking in consideration to find intermediate values between 0 and 80% parasite death for BTZ.

Answer: Thanks for the comments and contribution. New assays were performed and the data included.

Careful revising Figures, Legends and Results: The text’s explanation of figures and legends does not match. In lines 314 to 316, the authors describe that Figure 2 is the “cytotoxic profile of BTZ on L929 cells”, implying an assay in non-infected cells. However, the legend in Fig 2 says: “against intracellular amastigotes of T. cruzi”.

Answer: Thanks. The text was revised.

In lines 316 to 320, the potency results against parasites are presented as illustrated in Fig 3. Nonetheless, the legend describes “L929 mammalian cell lines”. Considering the concentration values in the graph, it is possible to realize that the legends are misplaced. Please revise all information provided as a result from all the figures and tables.

Answer: Thanks. The text was revised.

Figure 6 was improved as requested. Please consider the following changes:

The X-axis should be changed to “day post-infection” since the authors refer to infection days as “dpi”. The symbol used to demonstrate the period of drug administration is called “dashed lines”; however, the legend is described as “Dot lines”.

Include in the legend the number of mice used in each group.

Answer: Thanks. The legend and figure 6 were revised.

Revise written English: comma in numbers instead of dots, units, subscripts, superscripts, and other minor issues.

Answer: Thanks. The text was revised.

---

## [Reviewer Report · REVIEWERS’ COMMENTS]

## REVIEWER #1

Reviewer comments: The manuscript is able to be accepted in its current form.

## REVIEWER #2

Reviewer comments: The addition of a new experiment and the modifications made in the text were appreciated. However, it was noticed that changes in the overall values or even changes in the standard deviation of IC50 in the text or tables were not performed.

Congratulations on the work, and I hope that the considerations made improve the scientific knowledge of the research group.